# SOXC are critical regulators of adult bone mass

Marco Angelozzi [1,2] ✉, Anirudha Karvande [1,2] & Véronique Lefebvre [1] ✉

Pivotal in many ways for human health, the control of adult bone mass is governed by complex, incompletely understood crosstalk namely between mesenchymal stem cells, osteoblasts and osteoclasts. The SOX4, SOX11 and SOX12 (SOXC) transcription factors were previously shown to control many developmental processes, including skeletogenesis, and *SOX4* was linked to osteoporosis, but how SOXC control adult bone mass remains unknown. Using SOXC loss- and gain-of-function mouse models, we show here that SOXC redundantly promote prepubertal cortical bone mass strengthening whereas only SOX4 mitigates adult trabecular bone mass accrual in early adulthood and subsequent maintenance. SOX4 favors bone resorption over formation by lowering osteoblastogenesis and increasing osteoclastogenesis. Single-cell transcriptomics reveals its prevalent expression in *Lepr*+ mesenchymal cells and ability to upregulate genes for prominent anti-osteoblastogenic and pro-osteoclastogenic factors, including interferon signaling-related chemokines, contributing to these adult stem cells' secretome. SOXC, with SOX4 pre-dominantly, are thus key regulators of adult bone mass.

Many types of diseases can affect adult bone mass and thereby affect various physiological processes and cause mobility restraint, pain, and mortality. At one end of the spectrum, osteoporosis, a prevalent bone fragility disorder, is characterized by extensive bone loss and micro-architectural changes that greatly increase fracture risk[1]. It develops earlier in women than men, due to postmenopausal estrogen depletion[2], but both sexes show early-onset idiopathic forms and increased prevalence with aging[3]. At the other end, osteopetrosis, featuring overly dense bones[4], predisposes to fractures too, and to extra-skeletal features, including neurologic manifestations and bone marrow failure. These diseases are caused by genetic variants and many other factors that typically create an imbalance between osteoblast-driven bone formation and osteoclast-driven bone resorption[5]. Various therapies are approved to promote balance restoration, but none is fully curative and devoid of grim side effects[6].

Numerous factors and signaling pathways have been described that control bone cell fate and activities. Master regulators of osteo-blastogenesis include the Runt-domain transcription factor RUNX2 and the SP1-family transcription factor SP7[7], while master regulators of osteoclastogenesis include the ETS-family transcription factor PU.1 and the nuclear-factor-of-activated-T-cell-family transcription factor NFATC1[8]. Sex hormones, parathyroid hormone, leptin, and inflammatory and nerve signals are some of the key endocrine and paracrine factors controlling bone mass[9–12]. They originate from multiple cell types and decisively influence the crosstalk between osteoblasts and osteoclasts. Within the bone environment, these cells include endo-thelial cells[13], adipocytes[14], bone marrow mesenchymal stem cells (MSCs)[15], and immune cells[16]. Despite the current state of advanced knowledge, many questions remain open, hampering the design of better treatments for bone diseases.

SOX4, SOX11, and SOX12 form the SOXC group in a family of transcription factors (TFs) characterized by a DNA-binding domain highly similar to that of SRY (sex-determining-region-on-the-Y-chro-mosome). SOXC proteins are closer to one another than to other relatives in this domain and they also feature a group-specific trans-activation domain. Most SOX proteins work in concert with other TFs to govern lineage commitment and differentiation of discrete cell types, such that, as an entity, the SOX family controls almost every cell

[1]Department of Surgery, Division of Orthopaedics, Children's Hospital of Philadelphia, Philadelphia, PA, USA. [2]These authors contributed equally: Marco Angelozzi, Anirudha Karvande. ✉e-mail: angelozzim@chop.edu; lefebvrev1@chop.edu

lineage[17]. Besides encoding similar proteins, the SOXC genes are also largely co-expressed in mesenchymal, neuronal, and other progenitor cell types from organogenesis onwards[18–20]. It was shown, mainly through loss-of-function mouse models, that SOX4 and SOX11 work largely redundantly and SOX12 minimally in the control of cell survival, proliferation, and differentiation. Notably, SOXC deletion in embryonic skeletogenic mesenchyme led to major cartilage primordia patterning defects and to an absence of joint, growth plate, and endochondral bone development[21–23]. In humans, *SOX4* and *SOX11* heterozygous loss-of-function variants have been associated with neurodevelopmental syndromes characterized by intellectual disability, mild dysmorphism, digit anomalies, and various inconsistent manifestations[24–26]. Interestingly, a genome-wide association study identified *SOX4* among the most significant genes associated with hip bone mineral density (BMD) in postmenopausal women[27]; single-nucleotide polymorphisms (SNPs) in the *SOX4* 3′UTR were associated with osteoporosis[28]; and *SOX4* expression was found to be reduced in bone biopsies of osteoporotic women[29]. Neither *SOX11* nor *SOX12* has been linked to adult bone diseases. In line with these findings, *Sox4* heterozygous-null mice were found to be mildly osteopenic[30]. Their femurs had a thinner bone cortex, thinner but slightly more bone trabeculae, and weaker mechanical properties. In vitro data suggested that SOX4 stimulates osteoprogenitor proliferation and differentiation by upregulating *Sp7* and SP7/RUNX2 target genes, such as *Alpl* (encoding alkaline phosphatase) and *Bglap* (encoding osteocalcin)[30]. Similarly, another in vitro study suggested that SOX11 promotes the expression of *Runx2* and *Sp7* in osteogenic cells[31]. However, these roles have not been validated in vivo yet.

In this work, we analyze mice with SOXC loss or gain of function in osteogenic cells to test if and how SOXC control adult bone mass accrual and maintenance. We show that the three genes promote cortical bone thickening and widening in adolescence and that SOX4 mitigates trabecular mass accrual and maintenance in adulthood. SOX4 delays osteoblastogenesis and promotes osteoclastogenesis, namely by controlling the expression of key secretome components of *Lepr*+ MSCs.

## Results

### SOXC loss-of-function mouse model
The main mouse model that we used in this study combined conditional null alleles for all three SOXC genes and a tetracycline-repressed Cre transgene primarily active in the osteoblast lineage (*OsxCre*) (Fig. 1a). When conceived and raised in cages containing regular chow diet, most *SOXC^OsxCre^* (*Sox4^fl/fl^Sox11^fl/fl^Sox12^fl/fl^OsxCre*) mice were viable but underweight at weaning age, and *WT^OsxCre^* mice (SOXC wild-type alleles and *OsxCre*) were normal (Supplementary Fig. 1a). Unless otherwise stated, we silenced *OsxCre* in experimental mice until P21 by providing mice with food pellets supplemented with doxycycline (DOX, tetracycline antibiotic). Under this condition, *SOXC^OsxCre^* males and females still showed a mild weight deficit but looked otherwise healthy (Supplementary Fig. 1a–c). We analyzed their bone phenotypes at increasing ages, including pubescence (5 weeks), sexual maturity (7 weeks), early adulthood (10 weeks), skeletal maturity (13 weeks) and early aging (52 weeks).

### SOXC promote prepubertal accrual of cortical bone
Microcomputed tomography (μCT) of femurs revealed that the long bones of 5-week-old *SOXC^OsxCre^* male and female mice had a smaller cortical area (78% and 83%, respectively) than those of sex-matched controls and that this difference was largely maintained in adulthood (7–52 weeks; averages of 86–88%) (Fig. 1b; Supplementary Fig. 2a). This phenotype was due to reduced cortical thickness (average of 92%) and diameter (marrow area average of 71–90%). The BMD was normal in mutant mice at pubescence, but slightly reduced in adulthood (average of 97% by 52 weeks). Since *OsxCre* itself weakens cortical bones in

young mice[32], we tested *WT^OsxCre^* mice. These mice had lower bone parameters in the absence of DOX treatment, but not when they were raised with DOX (Supplementary Fig. 2b, c). Thus, the cortical bone phenotype of *SOXC^OsxCre^* mice is due to SOXC loss.

Histomorphometry analysis of cortical bone endosteal surfaces provided a cellular explanation for this cortical bone phenotype, its early onset, and its stability in adulthood. It showed indeed that osteoblasts were present in normal numbers in *SOXC^OsxCre^* mice at all ages (Fig. 1c), but formed and mineralized bone less actively in pubescence (Fig.1d, e). Osteoclasts were unlikely involved in this phenotype since they were virtually absent in young mice and present in normal numbers in adults (Supplementary Fig. 2d).

Three-point bending tests showed that SOXC inactivation reduced the biomechanical properties of femurs in females, but not males (Fig. 1f, g). Indeed, femurs, which were 20% less stiff in control females than males at 13 weeks, saw their stiffness slightly more reduced upon SOXC inactivation in females (84%) than males (89%), such that they were 27% less stiff in mutant females than males. Moreover, femurs were 29% less resistant to load in 13-week-old control females than males and SOXC inactivation increased this difference to 49%. However, control and mutant bones were equally stiff and resistant to load in 52-week-old males and females. Bone elasticity, as assessed by the Young's modulus, was higher in female than male bones at 13 and 52 weeks and tended to increase in all mice upon SOXC inactivation. These differences were not due to *OsxCre* itself (Supplementary Fig. 2e).

We concluded that SOXC help widen long bone diaphyses and thicken their cortex in juvenile mice, regardless of sex. They also increase the mechanical properties of female bones. Their impact persists through adulthood, where they also help increase BMD.

### SOXC lower trabecular bone mass accrual and maintenance
μCT of the femur secondary spongiosa showed that *SOXC^OsxCre^* males and females had a normal trabecular bone mass (bone volume/tissue volume, BV/TV) at pubescence (5 weeks), but the thickness of trabeculae, like that of cortical bone, was reduced (Fig. 2a, b). Interestingly, both males and females reached a BV/TV higher than that of control mice in early adulthood (10 weeks; 1.3- and 1.5-fold, respectively). Subsequently, however, mutant males largely lost this advantage (1.1-fold at 52 weeks), while mutant females further increased it (1.8-fold at 52 weeks). These findings were confirmed on tissue sections (Fig. 2c). The progressive increase in BV/TV seen in mutant females was mainly due to the maintenance of a high number of trabeculae throughout adulthood. Indeed, while control males and females steadily reduced their trabecular number between 5 and 52 weeks (2.6- and 2.1-fold, respectively), mutant males reduced it less (1.9-fold), and mutant females kept it intact (1.0-fold). In contrast, while trabecular thickness steadily increased in control males and females throughout adulthood (1.4- and 1.5-fold, respectively, by 52 weeks), it increased in mutant males in early adulthood and decreased thereafter (1.1-fold by 52 weeks) and it barely increased in mutant females (1.2-fold). Trabecular BMD increased similarly in mutant (10%) and control (11%) males between 5 and 52 weeks but increased less in mutant (9%) than control (16%) females. Also, mutant males and females maintained a higher connectivity density of the trabecular network than control mice in adulthood, and mutant females increased their number of plate-like structures, as indicated by a lower structural model index (Supplementary Fig. 3a). The analysis of the trabecular bone phenotype of *WT^OsxCre^* mice verified that the trabecular bone gain of *SOXC^OsxCre^* mice was mainly caused by SOXC inactivation (Supplementary Fig. 3b). Noticeably too, *SOXC^OsxCre^* mice developed the same trabecular bone phenotype between 5 and 13 weeks when they were raised without DOX food, indicating that the increase in trabecular bone mass was truly linked to puberty (Supplementary Fig. 3c).

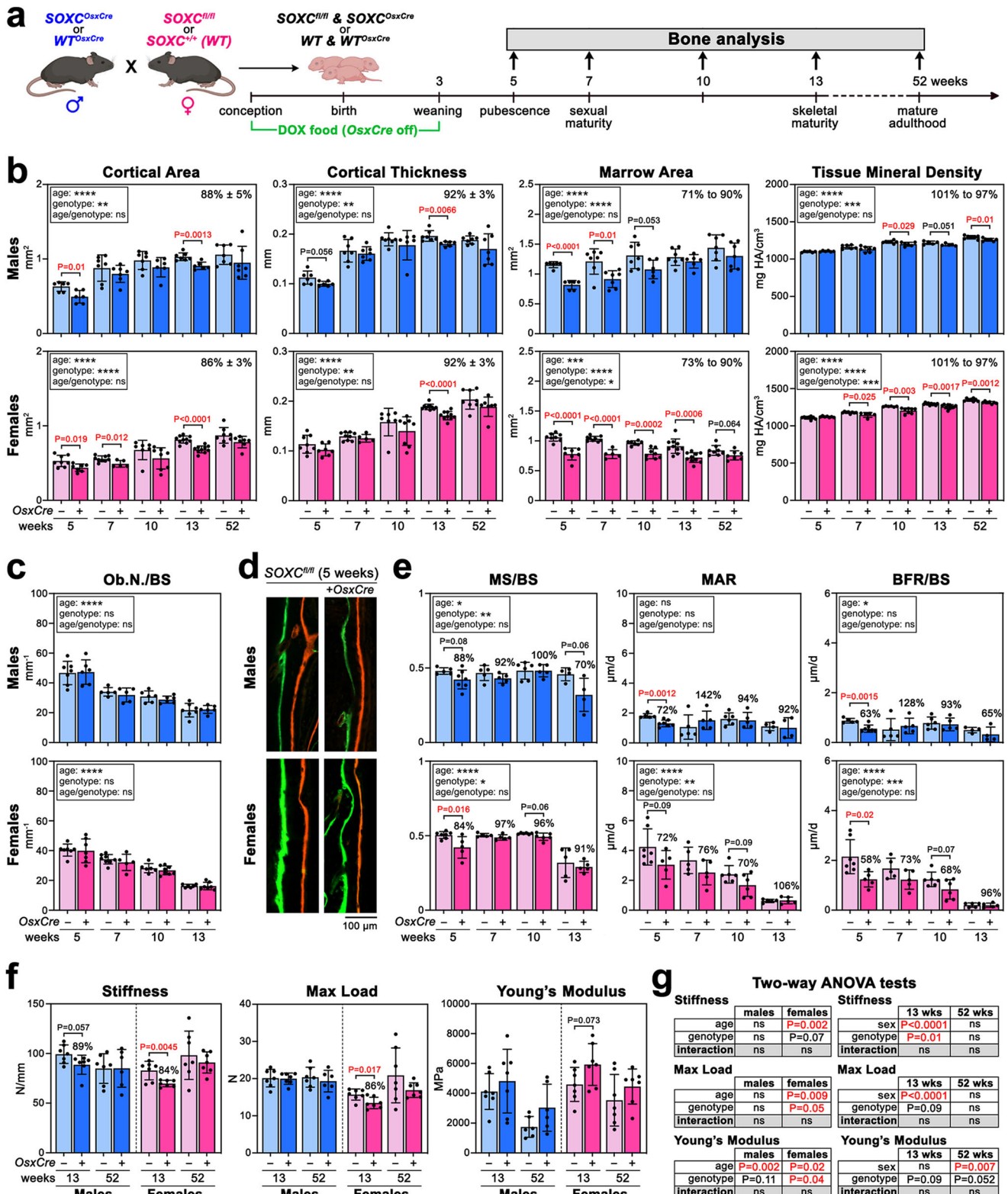

Static histomorphometry showed that *SOXC^OsxCre* females increased their osteoblast number upon reaching sexual maturation (7 weeks), just before extending the bone surface, and then maintained osteoblast numbers in proportion to the trabecular area (Fig. 2d, e). Dynamic histomorphometry showed that the surge in osteoblast number occurring at puberty matched an increase in bone formation rate and that *SOXC^OsxCre* females had an elevated mineralizing surface at all ages, but a reduced mineralization rate (Fig. 2f, g). This may explain

why they maintained a high number of trabeculae, but barely thickened them. Mutant females had fewer osteoclasts per bone surface than control mice by 10 weeks and maintained this deficiency afterwards (Fig. 2h, i). Serum analyses showed a higher ratio of P1NP (N-terminal propeptide of type I procollagen, marker of bone formation) to CTX-1 (crosslinked C-terminal telopeptide of type I collagen, marker of bone resorption) in mutant versus control females at 7 weeks, matching the increase in bone formation rate; and a lower ratio of

**Fig. 1 | Experimental design and cortical bone phenotype of *SOXC^OsxCre* mice.**
**a** Experimental design for the generation and bone analysis of *SOXC^OsxCre* and control mice (image created with BioRender.com). **b** Quantification of cortical area and thickness, marrow area, and cortical BMD of femurs obtained by µCT analysis of *SOXC^OsxCre* and control males and females at 5–52 weeks. Each dot corresponds to a distinct mouse. Bars and brackets represent means and standard deviations, respectively. Statistical differences were assessed by two-sided unpaired Student's *t*-tests. *P*-values lower (red) and near (black) 0.05 are indicated. Box, statistical differences obtained using a two-way ANOVA test of the effects of age and genotype and their interaction (ns, non-significant; *$p \leq 0.05$; **$p \leq 0.01$; ***$p \leq 0.001$; ****$p \leq 0.0001$). The percentage averages of mutant versus control values, all ages combined, or their ranges are indicated along with their standard deviations. **c** Static histomorphometry of osteoblast numbers per bone surface (Ob.N./BS) at the endosteal surfaces of femurs from *SOXC^OsxCre* and control males and females at

5–13 weeks. Data are presented as in (**b**). No significant difference was detected between *SOXC^OsxCre* and control mice. **d** Representative pictures of in vivo labeling of newly synthesized mineralized matrix at the endosteal surfaces of femurs from 5-week-old *SOXC^OsxCre* and control males and females. Mice were injected with calcein (green) and alizarin red (red) nine and two days before euthanasia, respectively. The assay was conducted for all the mice analyzed in (**e**). **e** Dynamic histomorphometry of the mineralizing surface (MS/BS), mineral apposition rate (MAR), and bone formation rate (BFR/BS) per bone surface of endosteal surfaces of femurs from *SOXC^OsxCre* and control males and females at 5–13 weeks. Data are presented as described in (**b**). **f** Biomechanical properties of femurs from *SOXC^OsxCre* and control males and females at 13 and 52 weeks determined in three-point bending tests. Data are presented as described in (**b**). **g** Results of two-way ANOVA tests using data in (**f**) to assess the effects of age, sex, genotype, and their interactions on the biomechanical properties of *SOXC^OsxCre* and control bones.

RANKL (receptor activator of nuclear-factor kB ligand, osteoclastogenesis activator) to OPG (osteoprotegerin, osteoclastogenesis inhibitor) in mature adults, matching differences in osteoclast numbers (Fig. 2j). Of note, the P1NP level was or tended to be elevated in mutant females at all ages, and the CTX-1 level was or tended to be increased in mutant females at 13 and 52 weeks, indicating that the coupling of the osteoblast and osteoclast activities was not fully disrupted (Supplementary Fig. 4a). The cortical bone phenotype of these mutants unlikely affected levels of serum markers significantly since it was stable by the time (5 weeks) of the analysis and since the adult turnover of trabecular bone is more intense than that of cortical bone[33,34]. *SOXC^OsxCre* males showed similar, but less marked changes (Supplementary Fig. 4b–i). *SOXC^OsxCre* females, and *SOXC^OsxCre* males to a lower extent, thus augmented their long bone trabecular bone mass by increasing their osteoblast number and activity from early adulthood onwards and by lowering their osteoclast number and activity later on.

Histomorphometry of lumbar vertebrae also showed an increase in BV/TV and trabecular number in 13- and 52-week-old *SOXC^OsxCre* mice, but differently from long bones, this increase was similar in males and females (Supplementary Fig. 5a, b). Males also showed a larger trabecular diameter.

We concluded that SOXC lessen adult trabecular bone mass accrual and subsequent maintenance in long bones and vertebrae of mice of both sexes. Their effect on long bones is transient in males but lasting in females.

## SOXC may impede bone mass control by female hormones
Since *SOXC^OsxCre* adult females developed a more pronounced phenotype than males in long bones and since sex hormones critically control trabecular bone mass[9], we asked whether SOXC may functionally interact with female sex hormones in this process. We ovariectomized 13-week-old females and analyzed bone parameters two months later (Supplementary Fig. 6a). Mutant mice lost a larger amount of trabecular bone than controls upon ovariectomy (BV/TV loss of 1.12 versus 0.07) (Supplementary Fig. 6b–d). They partially lost their trabecular number advantage (decrease from 1.7- to 1.4-fold), while control mice lost most of their trabecular thickness advantage (decrease from 1.3- to 1.1-fold). A significant interaction between SOXC and ovariectomy in determining trabecular number and thickness was validated by two-way ANOVA tests ($p = 0.007$ and 0.004, respectively). These data support the notion that SOXC may impede the beneficial actions of female sex hormones on trabecular bone mass accrual and maintenance.

## All SOXC control cortical bone and only SOX4 trabecular bone
We analyzed single *SOXC^OsxCre* mutants to evaluate the contribution of each SOXC gene to the control of adult bone mass. Cortical bone parameters were equally or less affected in single compared to triple mutants, suggesting redundant functions in bone cortex development (Supplementary Fig. 7). In contrast, *Sox4^OsxCre* mice increased their

trabecular bone mass in adulthood almost as much as *SOXC^OsxCre* mice, whereas *Sox11^OsxCre* and *Sox12^OsxCre* mice did not (Fig. 3a). Thus, only SOX4 regulates adult trabecular bone mass.

Single-cell RNA sequencing (scRNA-seq) of bone and marrow cells from young male mice[35] showed weak if any expression of *Sox11* and *Sox12* in the surveyed cell types (Fig. 3b, c). In contrast, *Sox4* was highly expressed in *Lepr*⁺ MSCs (also called adipoCARs[36] and MALPs[35]) and was expressed at a lower but substantial level in preosteoblasts (pre-OBs), mature osteoblasts (OBs), pericytes, and endothelial cells. As expected, *Sox4* was also expressed in hematopoietic progenitors[37] and B cells[38]. Accordingly, *Sox4* mRNA and protein were detected in situ in bone marrow cells, osteoblasts, and osteocytes (Fig. 3d, e). They were also seen in terminal growth plate chondrocytes (GPCs, not captured in scRNA-seq).

Since hypertrophic GPCs give rise to a subset of trabecular osteoblasts[39,40], and since *OsxCre* is weakly active in GPCs[41], we asked whether *Sox4* deletion in GPCs could contribute to the bone phenotype of *Sox4^OsxCre* mice. We generated females in which we inactivated *Sox4* in limb bud mesenchymal progenitors of osteoblasts, chondrocytes and other skeletal cells (*Sox4^Prx1Cre*), in chondrocyte-lineage cells (*Sox4^AcanCreER*, with tamoxifen-mediated Cre activation at P21), or in hypertrophic GPCs (*Sox4^Col10Cre*) (Fig. 3f, g). µCT of femurs showed that 13-week-old *Sox4^Prx1Cre* and *Sox4^OsxCre* females had the same trabecular bone phenotype. *Sox4^AcanCreER* and *Sox4^Col10Cre* females had a reduced BV/TV, as trabeculae were present in normal number but were as thin as in *Sox4^OsxCre* and *Sox4^Prx1Cre* mice. Thus, SOX4 may control trabecular thickness via roles in terminal chondrocytes or chondrocyte-derived osteoblasts, and trabecular number via roles in other osteoblast-lineage cells.

To consolidate these findings, we created a conditional SOX4 gain-of-function mouse model, where we knocked-in the human *SOX4* coding sequence (*hSOX4*) into the *Gt(ROSA)26Sor* locus (referred to as *R26^SOX4*) (Supplementary Fig. 8a–c). We expressed hSOX4 in limb skeletal cells using *Prx1Cre* (Supplementary Fig. 8d). *R26^SOX4/SOX4*Prx1Cre* females had slightly shorter growth plates and limbs than normal but looked otherwise healthy (Supplementary Fig. 8e–g). Interestingly, their femur cortical bone area and thickness were increased by 15% and 11%, respectively, compared to those of controls (Supplementary Fig. 8h). More strikingly, trabeculae were 44% less numerous and 36% thicker than in controls, resulting in a 57% lower trabecular mass (Fig. 3h). Thus, opposite to SOXC loss-of-function, SOX4 overexpression in skeletal cells promoted bone cortex and trabecular thickening, and drastically reduced trabecular number.

## SOXC primarily alter the transcriptomes of *Lepr*⁺ MSCs
To identify cellular and molecular targets of SOXC in the control of adult trabecular bone mass, we performed scRNA-seq assays. We collected femurs and tibiae from two pairs of 7- and 13-week-old *SOXC^OsxCre* and control sisters, thoroughly scraped the periosteum away, enzymatically digested the cortical bone, trabecular bone, and endosteal

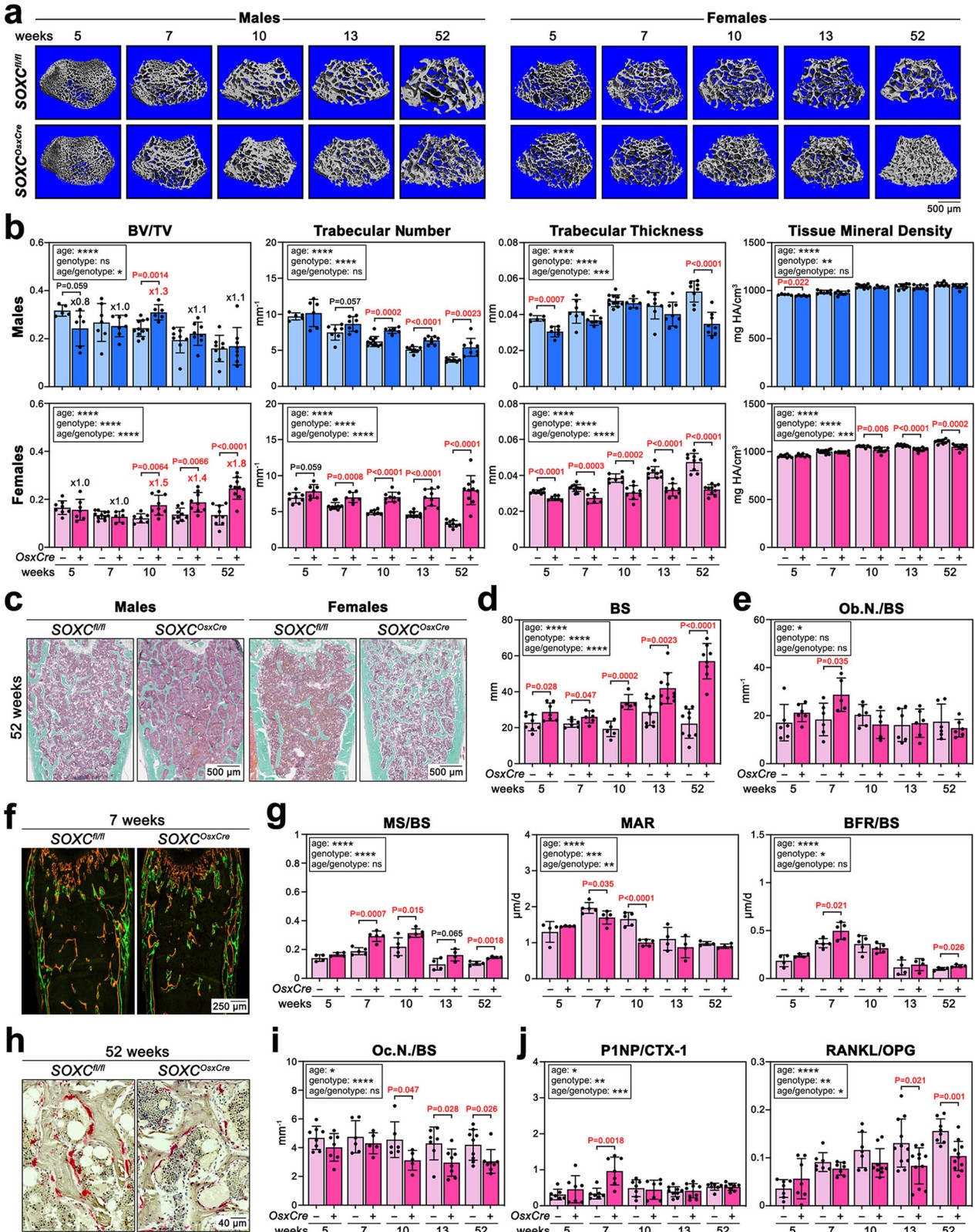

bone marrow tissues, and immunomagnetically sorted out hemato-poietic cells (Fig. 4a). After filtering out doublets and cells with low-quality transcriptomes, we analyzed $7696 \pm 500$ control and $7716 \pm 1084$ mutant cells per sample. Uniform manifold approximation and projection (UMAP) dimensional reduction identified 27 cell clusters (Supplementary Data 1a). Based on marker expression we regrouped them in 17 clusters corresponding to distinct lineages and

differentiation stages (C1–C17; Fig. 4b, c and Supplementary Data 1b). C1 contained $Sca1^+$ (also referred to as $Ly6a^+$) MSCs originating from bone marrow (also known as PDGFRA$^+$SCA1$^+$ or PαS MSCs[42] and as early mesenchymal progenitors or EMPs[35]) and possibly also from residual periosteum[43]. C2 and C3 featured $Lepr^+$ MSCs and $Lepr^{low}Runx2^{low}$ osteoprogenitors (OPs or osteoCARs[36]), respectively. C4 and C5 contained preosteoblasts (preOBs, $Runx2^{high}Sp7^{low}$), and

**Fig. 2 | Trabecular bone phenotype of *SOXC^OsxCre* mice at 5 to 52 weeks.**
**a** Representative μCT images of femur secondary spongiosa from *SOXC^OsxCre* and control males and females. **b** Quantification of BV/TV, trabecular number and thickness, and mineral density of femurs from the same mice as in (**a**). Each dot corresponds to a distinct mouse. Bars and brackets represent means and standard deviations, respectively. Statistical differences were assessed by two-sided unpaired Student's *t*-tests. *P*-values lower (red) and near (black) 0.05 are indicated. Box, statistical differences obtained using a two-way ANOVA test for the effects of age and genotype and their interaction (ns non-significant; *$p \leq 0.05$; **$p \leq 0.01$; ***$p \leq 0.001$; ****$p \leq 0.0001$). **c** Representative pictures of Masson–Goldner's trichrome stained femur sections from 52-week-old mice. Mineralized bone, green; osteoid tissue and marrow, pink/dark red; red blood cells, bright red. The assay was done for all mice analyzed in (**d**). **d** Static histomorphometry of femur trabecular bone surface (BS) from *SOXC^OsxCre* and control females. Data are presented as in (**b**). **e** Static histomorphometry of osteoblast

numbers per bone surface (Ob.N./BS) for *SOXC^OsxCre* and control females. Data are presented as in (**b**). **f** Representative pictures of in vivo labeling of newly synthesized mineralized matrix in trabecular bones from 7-week-old *SOXC^OsxCre* and control females. Mice were injected with calcein (green) and alizarin red (red) nine and two days before euthanasia, respectively. The assay was done for all mice analyzed in (**g**). **g** Dynamic histomorphometry of mineralizing surface (MS/BS), mineral apposition rate (MAR), and bone formation rate (BFR/BS) per femur bone surface. Data are presented as in (**b**). **h** Representative pictures of TRAP staining of sections through femur trabecular bone and marrow from 52-week-old females. TRAP⁺ cells, red; bone matrix, light brown. The assay was done for all mice analyzed in (**i**). **i** Static histomorphometry of osteoclast numbers (Oc.N./BS) per bone surface. Data are presented as in (**b**). **j** Serum level ratios of P1NP/CTX-1 (bone formation/resorption indicators) and RANKL/OPG (osteoclastogenesis activator/inhibitor). Data are presented as in (**b**).

C6 contained mature osteoblasts and osteocytes (OBs, *Sp7^high Bglap^high*). C7 cells were probably GPC-derived OBs (*Col2a1⁺Col10a1⁺Bglap⁺*). C8 and C9 were GPCs, with C8 expressing pre/hypertrophic markers (*Ihh⁺Col10a1⁺*). C10 featured pericytes (*Rgs5⁺*) and C11 featured endothelial cells (*Pecam1⁺*). C12 contained apoptotic cells and C13–C17 contained hematopoietic cells. Confirming published data, *Sox4* was expressed most highly in *Lepr⁺* MSCs (C2), followed by *Sca1⁺* MSCs (C1) and osteoblastic cells (C3–C6) (Fig. 4c). It was also expressed in other populations, including endothelial cells and pericytes. Neither *Sox11* nor *Sox12* was notably expressed in skeletal lineage cells.

The analysis of C1–C11 showed that the populations of proliferating preOBs (C5) and pericytes (C10) were smaller (40 and 38%, respectively) in mutant than control samples, but the relative sizes of other populations were unaffected (Fig. 4d). Unlike seen in trabecular bone histomorphometry, mutant samples did not have more OBs, possibly because bones were incompletely digested. *Sox4* ranked among the most downregulated genes in mutant MSCs and osteogenic clusters (C1–C6), indicating efficient inactivation (Fig. 4e; Supplementary Data 2). These clusters had high numbers of differentially expressed genes (DEGs; log₂ fold change ≥0.25 and adjusted *p*-value ≤ 0.05), proving their reliance on SOX4, and C2–C4 shared more DEGs with one another than with other clusters, indicating similar SOX4 activities (Fig. 4f, g).

In sum, SOXC significantly impact the transcriptomes of MSCs and osteoblastic cells.

## SOXC reinforce *Lepr⁺* MSC signatures modulating bone mass

We used the complementary Gene Ontology (GO) and single-cell regulatory network inference and clustering (SCENIC)[44] tools to identify the molecular pathways most affected in mutant *Lepr⁺* MSCs and closely related *Lepr⁺* OPs and preOBs (C2–C4). Since DEGs were similar for 7- and 13-week-old mice (Supplementary Fig. 9a), we combined them for these analyses. GO analyses showed that both up- and down-regulated genes most significantly matched pathways associated with bone formation and wound healing (Fig. 5a; Supplementary Data 3a). In addition, downregulated genes significantly matched, but to a lesser extent, pathways involved in leukocyte and granulocyte chemotaxis and response to virus and interferons (IFNs). SCENIC analyses of the control C2–C4 transcriptomes identified 109 regulons, i.e., sets of genes predicted to be controlled by the same TF (Fig. 5b; Supplementary Data 4). A SOX4 regulon and 45 others were highly enriched in C2 *Lepr⁺* MSCs (C2R group), while 46 other regulons were more specific to C3 *Lepr⁺* OPs (C3R group) and 17 to C4 preOBs (C4R group). The C2R group contained the highest number of regulons affected in mutant cells. The SOX4 regulon had 27% of its genes downregulated upon SOXC inactivation and its overall activity significantly reduced (Fig. 5b, c; Supplementary Data 4). In agreement with GO analysis, regulons for TFs involved in interferon signaling and bone metabolism

had a high proportion of their genes downregulated and their overall activity significantly reduced in mutant cells. These regulons included those for STAT1 and STAT2 (signal transducer and activator of transcription 1 and 2), interferon regulatory factors (IRF2, IRF7, IRF8, and IRF9), and the nuclear-factor kappa subunit RELB. In contrast, upregulated genes and increased activities were found in the regulons of TFs promoting osteoblastogenesis and bone formation, such as the AP-1 subunits JUNB and FOS, CEBPD (CCAAT enhancer-binding protein delta, a RUNX2 co-factor), MEF2C and SP7 (Fig. 5b, c; Supplementary Data 4).

Many top DEGs belonging to GO-defined bone formation and wound healing pathways encode secreted factors (Fig. 5d; Supplementary Data 2, 3a). Downregulated ones included genes for factors with anti-osteoblastogenic and pro-osteoclastogenic activities, such as midkine (*Mdk*[45]), follistatin-like-1 (*Fstl1*[46]), secreted frizzled-related proteins (*Sfrp1* and *Sfrp2*[47,48]), beta-catenin interacting protein 1 (*Ctnnbip1*[49]), spondin-1 (*Spon1*[50]), chordin like-1 (*Chrdl1*[51]), pentraxin-3 (*Ptx3*[52]), and the vascular endothelial growth factor-A (*Vegfa*[53]). Interestingly, several of the most downregulated genes matching bone formation and wound healing pathways were predicted by SCENIC to be direct targets of SOX4 (Fig. 5e; Supplementary Data 4). These genes included *Mdk*, *Sfrp1*, *Sfrp2*, *Ctnnbip1*, and *Spon1*. Similar expression changes were observed for these gene sets in other clusters, such that the scores obtained for modules that combined these gene expression changes were proportional to *Sox4* expression and thus further support the prediction that several of these genes may be direct targets of SOX4 (Fig. 5f; Supplementary Fig. 9b and Supplementary Data 2).

Unlike downregulated genes, upregulated genes included many genes for pro-osteoblastogenic and anti-osteoclastogenic factors (Fig. 5d; Supplementary Data 2, 3a), such as cystatin C (*Cst3*[54,55]), WNT4[56], BMP6[57], and MMP13[58]. They also included genes for non-collagenous ECM proteins modulating bone formation and mineralization, such as osteocalcin (*Bglap*[59]), osteopontin (*Spp1*[60]), matrix gla protein (*Mgp*[61]), and CCN proteins (*Ccn1*, *Ccn2*, and *Ccn5*[62]). Several genes for important bone-related transmembrane enzymes also had their expression changed. *Gdpd2*[63] (osteoblast differentiation promoter glycerophosphodiester phosphodiesterase-3) and *Enpp1*[64] (mineralization inhibitor ectonucleotide pyrophosphatase/phosphodiesterase-1) were upregulated, whereas *Phex*[65] (pro-mineralization phosphate-regulating endopeptidase X-linked factor) was downregulated.

RT-qPCR for bone and marrow cells collected in the same way as for scRNA-seq validated many top DEGs in *SOXC^OsxCre* populations, including *Mdk* and *Sfrp1* downregulation and *Cst3*, *Bglap* and *Spp1* upregulation (Supplementary Fig. 9c). Mutant male cells also down-regulated *Mdk* and *Sfrp1*, strongly expressed in *Lepr⁺* MSCs, but did not upregulate *Cst3* and *Bglap*, strongly expressed in osteoblastic cells. This apparent sex bias in gene expression changes is consistent with the sex differences observed in osteoblast numbers and activities.

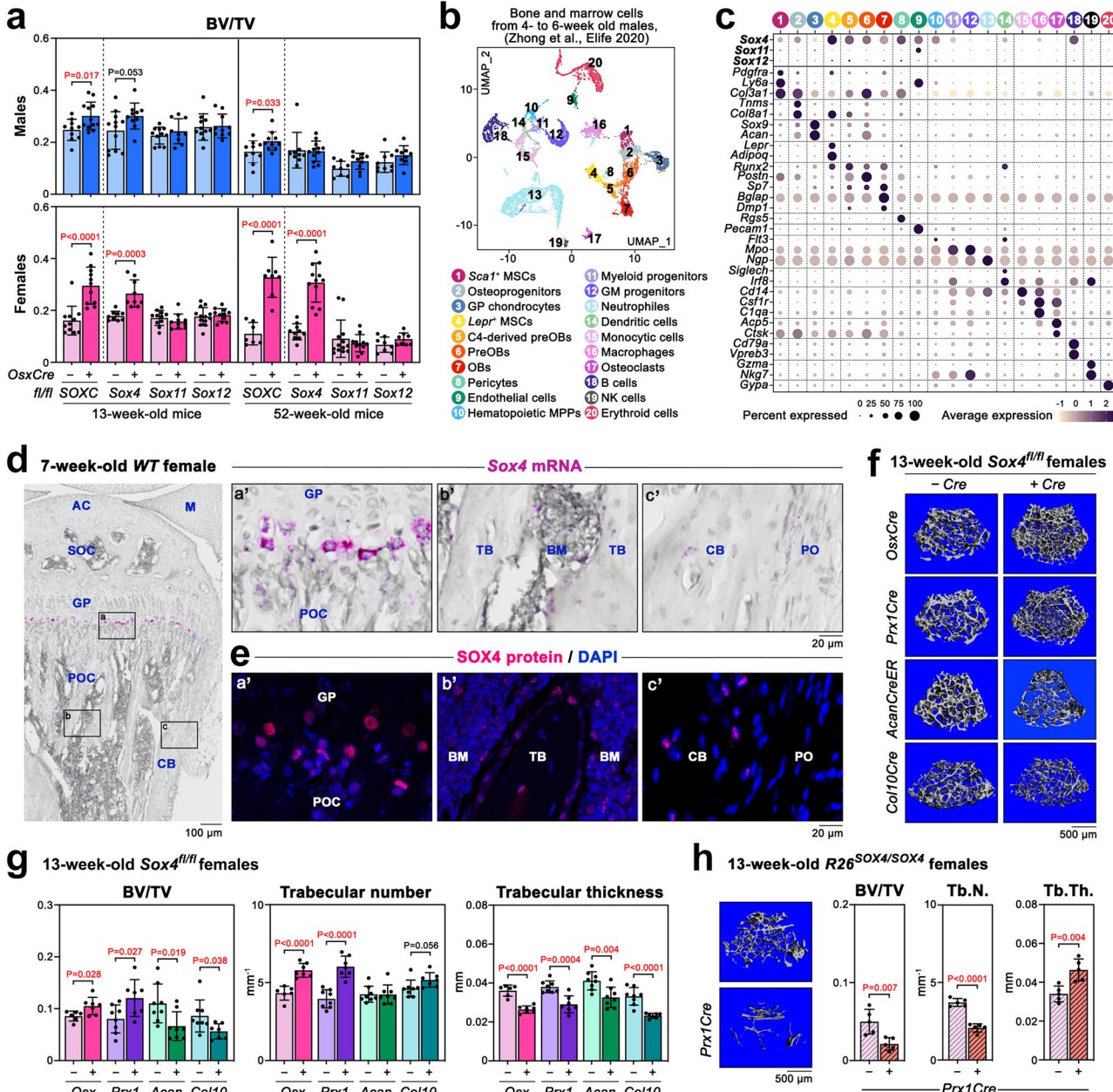

**Fig. 3 | SOX4 is the main SOXC controlling trabecular bone mass. a** μCT quantification of the BV/TV of femurs from $SOXC^{OsxCre}$, $Sox4^{OsxCre}$, $Sox11^{OsxCre}$, $Sox12^{OsxCre}$, and respective control males and females at 13 and 52 weeks. Each dot corresponds to a distinct mouse. Bars and brackets represent means and standard deviations, respectively. Statistical differences were assessed by two-sided unpaired Student's *t*-tests. *P*-values lower (red) and near (black) 0.05 are indicated. **b** UMAP plot of bone and bone marrow cell populations from 4- to 6-week-old male mice analyzed by scRNA-seq[35]. Twenty cell clusters were recognized, as indicated. **c** Dot plot showing the expression levels of the SOXC genes and major cell type markers in the bone and bone marrow cell clusters shown in (**b**). **d** *Sox4* RNA in situ hybridization of a section through the proximal tibia region of a 7-week-old female mouse. High-magnification images of highlighted areas are shown on the right: a', endochondral ossification front, including SOX4+ terminally differentiated GPCs; b', trabecular bone region; c', periosteum, and cortical bone. The magenta color representing

RNA signal was saturated, and the blue color resulting from counterstaining with hematoxylin was desaturated using Adobe Photoshop. AC articular cartilage, BM bone marrow, CB cortical bone, GP growth plate, M meniscus, PO periosteum, POC primary ossification center, SOC secondary ossification center, TB trabecular bone. The assay was repeated with 3 independent samples. **e** SOX4 immunostaining (red) in similar tissue regions as in (**d**, a'–c'). Nuclei are counterstained with DAPI (blue). The assay was repeated with 3 independent samples. **f** Representative μCT images of femur secondary spongiosa from 13-week-old $Sox4^{OsxCre}$, $Sox4^{Prx1Cre}$, $Sox4^{AcanCreER}$, $Sox4^{Col10Cre}$ and control females. **g** μCT quantification of BV/TV, trabecular number, and trabecular thickness in femur secondary spongiosa from same mice as in (**f**). Data are presented as in (**a**). **h** Representative μCT images of femur secondary spongiosa from 13-week-old $R26^{SOX4/SOX4}Prx1Cre$ and control females, and μCT quantification of BV/TV, trabecular number, and trabecular thickness. Data are presented as in (**a**).

Cells from $R26^{SOX4/SOX4}Prx1Cre$ females showed opposite changes, including *Mdk* upregulation and *Cst3* and *Spp1* downregulation (Supplementary Fig. 9d).

DEGs belonging to GO-defined IFN-related pathways included *Stat1*, *Jak2* (Janus kinase 2) and *Irf7*, key mediators of IFN signaling[66]; *Ccl2*, *Ccl19*, *Cxcl9*, *Cxcl10*, and *Cxcl13*, encoding chemokines regulating osteoclast precursor migration, recruitment and differentiation[67]; and other IFN-dependent genes, such as *Isg15* (ubiquitin-like protein) and *Gbp2*, *Gbp2b*, and *Gbp7* (guanine-binding proteins) (Fig. 5g; Supplementary Data 2). These genes were more affected at 13 than 7 weeks.

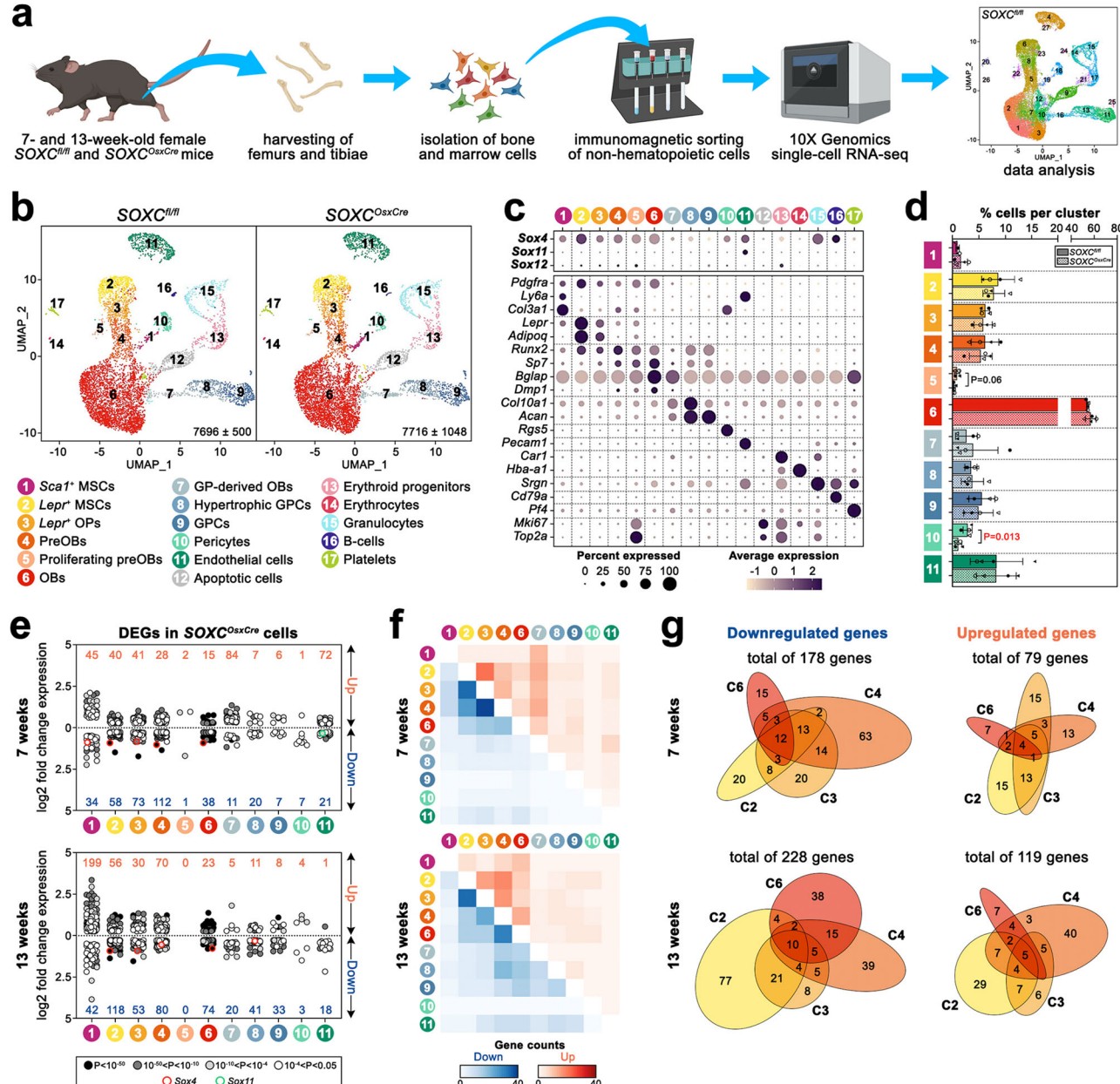

**Fig. 4 | scRNA-seq reveals transcriptomic changes in SOXC^OsxCre osteogenic cells.**
**a** Experimental strategy used to obtain suspensions of non-hematopoietic cells from cortical and trabecular bone, and endosteal bone marrow for scRNA-seq assays (image created with BioRender.com). **b** UMAP plots of bone and bone marrow cells from SOXC^OsxCre and control females (n = 4; 2 per genotype at 7 and 13 weeks). The average number of cells analyzed per sample is indicated with standard deviation. Seventeen distinct cell populations were identified as listed. **c** Dot plot showing the expression levels of the SOXC genes and major cell type markers used to identify clusters in (**b**). **d** Relative proportions of non-hematopoietic clusters in SOXC^OsxCre and control samples. Circles and triangles correspond to 7- and 13-week-old mice, respectively. Bars and brackets represent means and standard deviations, respectively. Statistical differences were assessed by two-sided paired Student's t-tests. P-values lower (red) and near (black) 0.05 are indicated. **e** Dot plots of the expression fold-changes of DEGs in mutant versus control populations at each age. The total numbers of up- and downregulated genes are indicated in orange and blue, respectively. The dot darkness reflects the significance of fold changes. Dots outlined in red and green represent Sox4 and Sox11, respectively. **f** Heatmaps showing the overlap of up- (orange) and down-regulated (blue) genes in mutant clusters. C5 was not included because it was too small to provide significant data. **g** Venn diagrams showing the numbers and overlap of genes down- or upregulated in C2–C4 and C6 at 7 and 13 weeks.

RT-qPCR for Stat1, Isg15, Ccl2, Cxcl9 and Cxcl10 in cells sorted in the same way as for scRNA-seq showed that each gene had its expression significantly (p ≤ 0.05) or nearly significantly (p ≤ 0.1) downregulated in mutant cells from females at or beyond 26 weeks and was not or less downregulated in mutant cells from 7-week-old females (Supplementary Fig. 9e). Mutant male cells showed similar gene expression changes but with lower amplitudes, in line with the milder trabecular bone phenotypes of males compared to females. Experiments performed in C3H/10T1/2 cells helped us consolidate the notion that SOX4 is involved in the activation of the interferon (IFN) pathway. These multipotent mesenchymal cells[68] highly expressed Sox4 and genes for interferon type I receptors, but neither Sox11 nor Sox12 (Supplementary Fig. 10a). As predicted, IFN-β robustly upregulated all tested genes in control cells, but much less when Sox4 was silenced (Supplementary Fig. 10b–d). Interestingly, none of these genes were predicted by SCENIC to be direct targets of SOX4 (Supplementary Data 4), but an

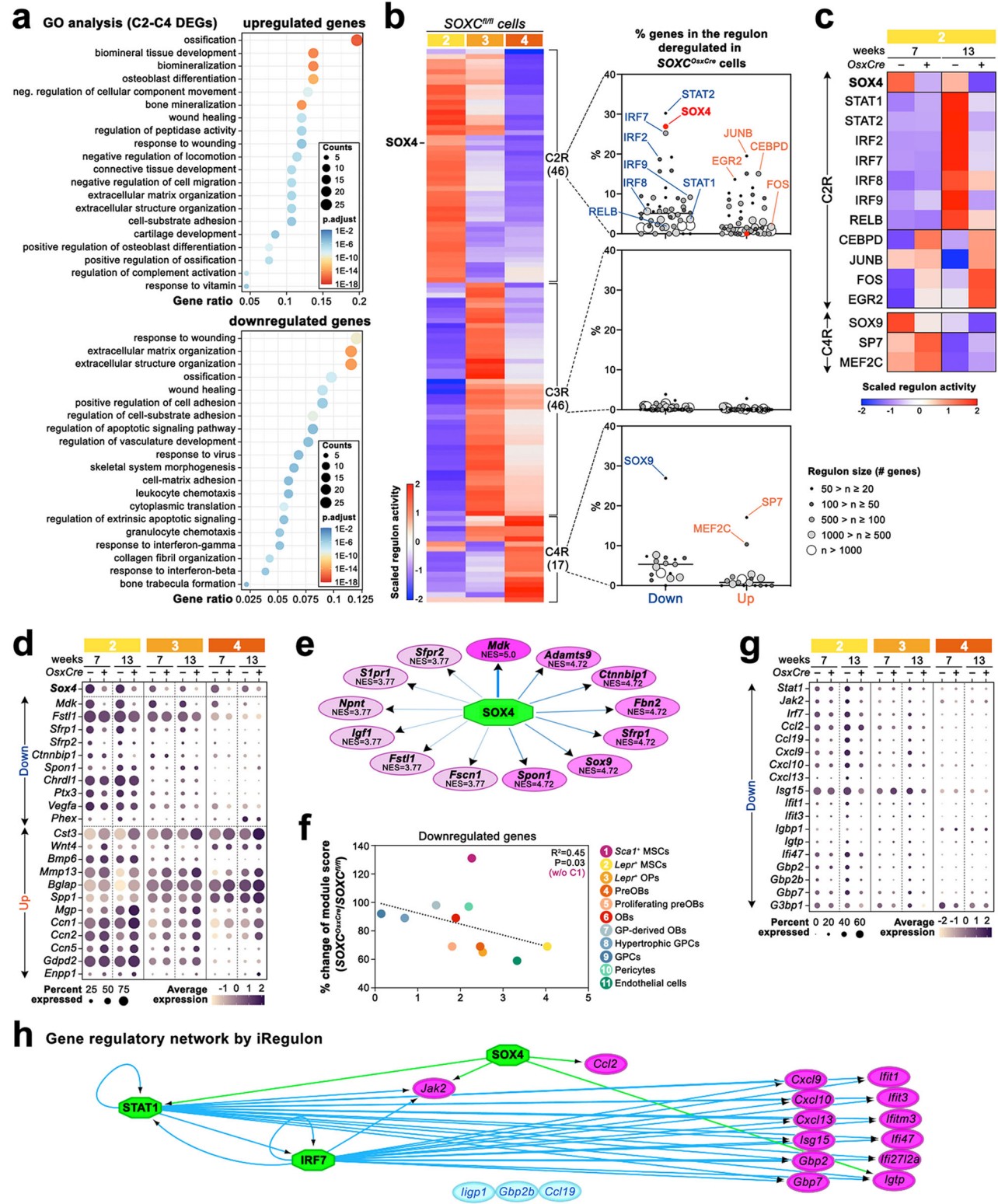

extended analysis of the gene regulatory network (GRN) with iRegulon[69] suggested that SOX4 may modulate the expression of IFN-dependent genes by promoting the expression of STAT1 and JAK2 (Fig. 5h).

Altogether, these results uncovered that SOX4 may directly activate anti-osteoblastogenic and pro-osteoclastogenic genes to delay osteoblastogenesis and facilitate bone mineralization and osteoclastogenesis. It may also promote the response to interferons by upregulating expression of key transcriptional mediators of the pathway.

## SOXC delay osteoblastic differentiation of bone marrow MSCs

To further investigate the notion that SOXC may delay the differentiation of OBs from MSCs, we used bioinformatics tools that help predict cell fate trajectories. Slingshot analysis of our scRNA-seq data for control and mutant cell populations at 7 and 13 weeks predicted that two cell lineages likely give rise to osteoblasts (Fig. 6a). One lineage would originate from *Lepr*+ MSCs/OPs (C2 and C3) and the other from *Sca1*+ MSCs (C1). Both MSC/progenitor types would give rise to preOBs (C4), which would then proliferate (C5) before maturing

**Fig. 5 | SOXC promote expression of anti-osteoblastogenic and pro-osteoclastogenic genes. a** GO enrichment analyses for biological processes using all genes up- or downregulated in C2–C4 mutant clusters at 7 and 13 weeks. *P*-values were calculated by hypergeometric distribution followed by Benjamini–Hochberg correction. **b** Heatmap of the activity scores of regulons identified by SCENIC in C2–C4 control cells, and dot plots of the percentage of genes in the regulons that were down- or upregulated in C2–C4 mutant cells. The dot area and color reflect the size of each regulon. SOX4 and most relevant regulons are indicated. **c** Heatmap of the activity scores of the most affected regulons in C2 mutant cells at 7 and 13 weeks. **d** Dot plots of the expression levels of genes selected as matching bone-related GO terms and being among the most differentially expressed ones in C2–C4 mutant clusters. **e** Graphic representation of the predicted target genes of SOX4 by SCENIC which are downregulated in C2–C4 mutant cells and match bone-related

GO terms. Pink shade reflects the normalized enrichment score (NES). Darker shades indicate higher values. **f** Correlation between *Sox4* expression levels in control populations and percentages of module scores obtained for mutant relative to control populations and generated using downregulated genes in the C2–C4 mutant cells that match bone-related GO terms. The linear relationship obtained using all data (except C1) is shown with a dotted line, and its significance is indicated by its *p*-value and coefficient of determination ($R^2$). **g** Dot plots of the expression levels of genes selected as matching interferon-related GO terms and being among the most downregulated ones in C2–C4 mutant clusters. **h** Graphic representation of the gene regulatory network for genes matching interferon-related GO terms and downregulated in C2–C4 mutant clusters. The graph was generated by iRegulon. Green octagons, regulators; pink ovals, target genes; cyan ovals, genes not identified as targets of any regulator.

into OB/osteocytes (C6). Although control and mutant cells were predicted to follow identical trajectories, pseudotime distribution proposed that both types of mutant MSCs might proceed faster than control cells towards OB differentiation. In agreement with Slingshot, RNA velocity analysis, which predicts the step-by-step directionality of cell progression, also suggested a faster progression of SOXC-mutant MSCs/osteoprogenitors towards OB differentiation (Fig. 6b).

In agreement with these predictions, GO analysis performed with all the genes upregulated in SOXC-mutant *Sca1*[+] MSCs in our scRNA-seq assays showed that many of these genes matched pathways related to ECM organization, ossification, and osteoblast differentiation (Fig. 6c; Supplementary Data 3b). These genes primarily encoded collagens (1, 3, 5, 6, and 16) enriched in bone and other proteins produced by osteogenic cells, such as periostin (*Postn*), osteoglycin (*Ogn*), osteocrin (*Ostn*), asporin (*Aspn*), and osteonectin (*Sparc*) (Fig. 6d).

To functionally validate the impact of SOXC inactivation on osteoblast differentiation, we established primary cultures of SOXC-deficient bone marrow-derived MSCs, which primarily contained *Sca1*[+] cells (Supplementary Fig. 11), and we analyzed the expression of osteoblast-lineage markers in cells either amplified in growth medium or cultured in osteogenic medium. We observed that SOXC-deficient cells expressed osteogenic markers (*Runx2, Sp7, Col1a1, Col3a1, Postn*, and *Bglap*) at higher levels than control cells in both culture conditions, and also tended to produce a higher amount of mineralized ECM (Fig. 6e, f).

Altogether, these results support the notion that SOXC delay the osteogenic differentiation of mesenchymal progenitors in the bone marrow of adult mice.

## Discussion

This study uncovered SOXC as key players in the control of adult bone mass acquisition and maintenance. It showed that the three genes strengthen cortical bone structurally and biomechanically in juvenile mice. Moreover, SOX4 mitigates trabecular bone mass accrual in early adulthood and subsequent maintenance by modulating the balance between bone formation and bone resorption (Fig. 7). At the cellular and molecular levels, SOX4 delays osteoblastogenesis and promotes osteoclastogenesis primarily by controlling the expression of major components of the *Lepr*[+] MSC regulatory secretome and interferon-related pathways. Previously identified links between *SOX4* and BMD and bone fragility in older women strongly support the translational relevance of these findings.

Body shaping and protection critically rely on the abilities of cortical and trabecular bones to resist mechanical stress[70]. We found that SOXC control both tissue types, but differentially. The three SOXC act redundantly in juvenile mice to increase the width and cortical thickness of long bones, whereas SOX4 acts solo in adulthood to lessen trabecular bone mass. SOX4 reduces the numbers of trabeculae but, like SOXC for the cortex, it increases their thickness. SOXC/SOX4 also ensure optimal bone mineralization, providing a complementary mechanism for the reduced mechanical properties of mutant bones.

Similar phenotypes were reported in mice lacking one allele of *Sox4* globally[30]. Similar actions in humans could thus explain why *SOX4* was linked to osteoporosis in aging women[29].

Previous studies conducted in vitro proposed that SOX4[30] and SOX11[31] stimulate osteoblast differentiation and do this by upregulating *Runx2* and *Sp7* expression. However, our scRNA-seq performed here with cells freshly isolated from adult long bones and recently with embryonic calvarial cells[23] as well as our in vitro assays performed with MSCs indicated an opposite effect as SOXC-deficient cells upregulated osteoblast differentiation markers. This discrepancy may be explained by our earlier findings that SOXC promote embryonic limb bud and calvarial progenitor cell survival and proliferation in vivo and in vitro[18,23]. Slow amplification of SOXC-deficient populations in previous studies possibly delayed osteoblast differentiation, including *Runx2* and *Sp7* upregulation. Alternatively, SOXC may have context-dependent activities on osteoblast differentiation both in vivo and in vitro.

A key finding from the present study is that *Lepr*[+] MSCs highly express *Sox4* and use this TF to enhance the expression of multiple components of their secretome that decisively control osteoclastogenesis, osteoblastogenesis, and bone mineralization. This finding complements evidence that *Lepr*[+] MSCs are powerful regulators of various cell lineages in bone and bone marrow[35,71–74]. The fact that these cells develop upon adulthood[75,76] fits with the adult onset of the trabecular phenotype of SOXC-mutant mice. Factors belonging to the SOX4-dependent *Lepr*[+] MSC secretome include the growth factor MDK, the WNT inhibitor SFRP1, and the cytokine PTX3. Knockout mice for *Mdk* and *Sfrp1* have a trabecular bone phenotype similar to that of *SOX*[OsxCre] mice[45,47]. Additionally, MDK and PTX3 induce osteoclastogenesis in vitro[52,77]. Conversely, *Cst3* (protease inhibitor cystatin C), *Spp1* (secreted phosphoprotein-1), and *Wnt4* were among the most upregulated genes in *SOX*[OsxCre] mutant *Lepr*[+] MSCs. Cystatin C stimulates osteoblastogenesis in vitro by enhancing BMP2 signaling[54], and inhibits enzymatic activity of osteoclasts[55]. Its role in vivo remains unknown. *Spp1* knockout mice have normal bones, but cells from these mice form osteoclasts in culture more efficiently than control cells[78]. WNT4 promotes bone formation in vitro and in vivo and reduces osteoclast formation in aging mice[56,79]. Altogether, the expression changes of these factors and others thus likely cause the imbalance between bone formation and bone resorption that results in the trabecular bone phenotype of *SOX*[OsxCre] adult mice.

Other molecular changes likely relevant to the *SOX*[OsxCre] adult bone phenotype include the downregulation of genes for the interferon signaling mediators *Stat1* and *Irf7* and targets thereof, including multiple chemokines. The roles of interferons in bone turnover remain controversial, possibly context-dependent[80]. However, knockout mice for *Stat1*, which mediates IFN signaling but also other signaling pathways, showed increased bone mass[81]. Furthermore, downregulated chemokine genes included major regulators of immune cells and osteoclasts[67]. *CCL2* (also known as *MCP-1*, monocyte chemoattractant protein 1), is highly upregulated in osteoporotic woman femurs[82], and knockout mice for CCL2 or its receptor CCR2 have elevated trabecular

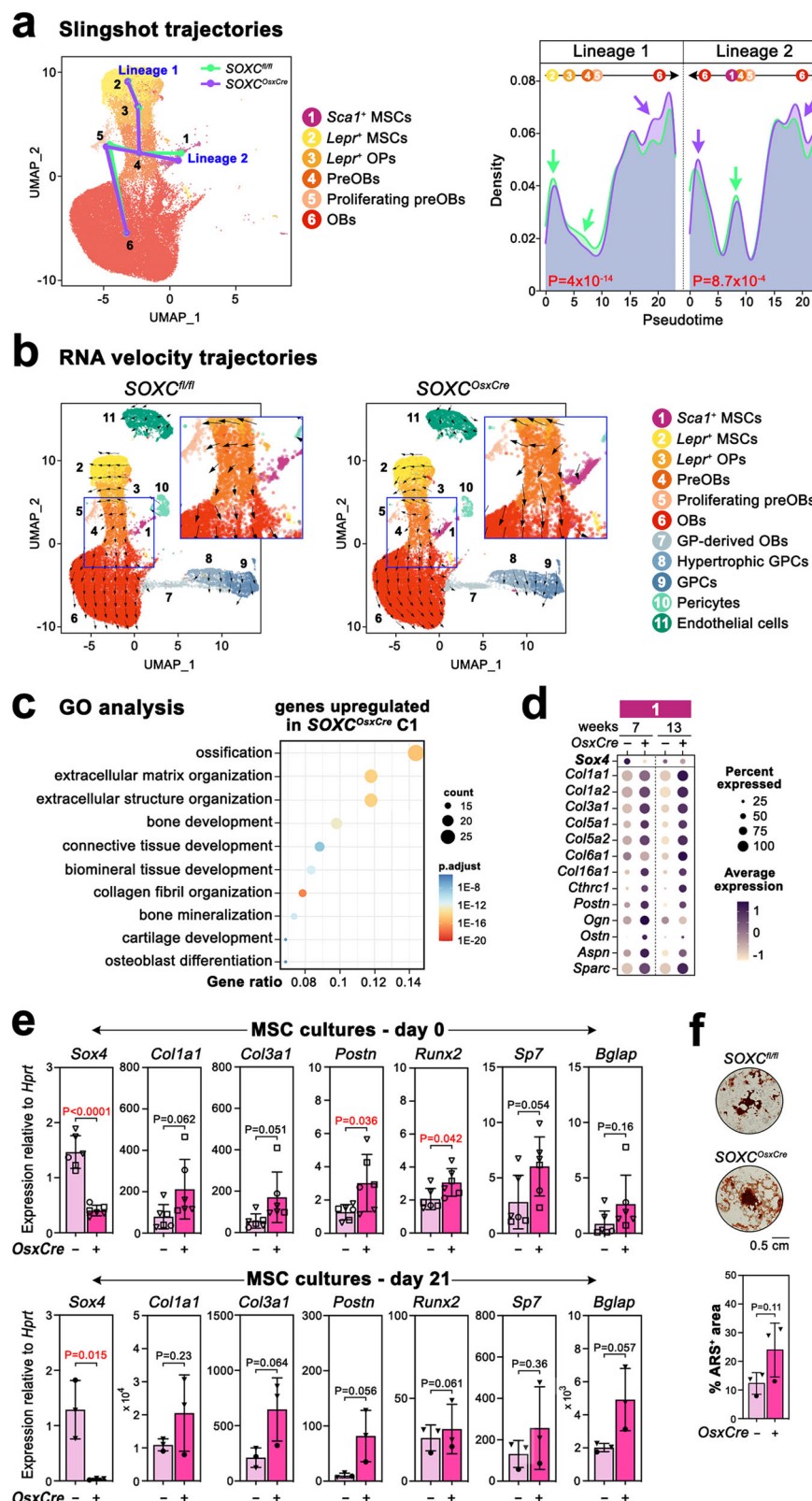

**a** Slingshot trajectories

1 *Sca1*⁺ MSCs
2 *Lepr*⁺ MSCs
3 *Lepr*⁺ OPs
4 PreOBs
5 Proliferating preOBs
6 OBs

**b** RNA velocity trajectories

1 *Sca1*⁺ MSCs
2 *Lepr*⁺ MSCs
3 *Lepr*⁺ OPs
4 PreOBs
5 Proliferating preOBs
6 OBs
7 GP-derived OBs
8 Hypertrophic GPCs
9 GPCs
10 Pericytes
11 Endothelial cells

**c** GO analysis — genes upregulated in *SOXC*^OsxCre C1

**d**

**e** MSC cultures - day 0 / MSC cultures - day 21

**f** *SOXC*^fl/fl / *SOXC*^OsxCre

bone mass[83,84]. CXCL9 and its receptor CXCR3 are involved in osteoclast progenitor recruitment to the bone matrix in fish[85], and both CXCL9 and CXCL10 support osteoclastogenesis in vitro[86,87]. These chemokines also affect other immune cells, such as T-cells, which can stimulate osteoclastogenesis and bone resorption in response to interferons[88]. The fact that the interferon pathway was mostly affected in older *SOXC*^OsxCre mice may explain why younger mutants had normal osteoclast numbers. This mechanism fits with the inflammaging theory[89] and thus makes SOX4 a candidate target for the prevention of age-related bone loss. Noteworthily, SOX4 is implicated in other pathologies linked to inflammaging, such as cancer[90], cardiovascular diseases[91], diabetes[92], and arthritis[93,94]. It will thus be worth assessing the presence of similar changes in these conditions to eventually propose multi-disease therapies.

**Fig. 6 | SOXC delay osteoblast differentiation. a** Trajectory analysis by Slingshot of *SOXC^OsxCre^* and control cells from 7- and 13-week-old females used in scRNA-seq assays. Predicted trajectories for control (green lines) and mutant (purple lines) cells mapped on a UMAP plot of all analyzed cells, and density plots of the pseudotime. Cluster labels are positioned based on their peak density distribution in the control plots. Of note, pseudotime analysis for lineage 2 suggests that C1 *Sca1^+^* MSCs could give rise to C6 OBs directly or via C4/C5 preOBs. **b** RNA velocity analysis of the same cells as in (**a**). UMAP plots show estimated velocities as black arrows. The arrow directions indicate the predicted directionality of cell lineage fate and differentiation. The arrow lengths depict the amplitudes of transcriptional changes. Blue square regions are magnified to highlight the main differences between control and mutant samples. The analysis of these regions predicts a faster progression of C1 *Sca1^+^* MSCs and C2/C3 *Lepr^+^* MSCs towards C6 OBs in mutant than control samples. **c** GO enrichment analysis of biological processes best matching the set of genes upregulated in C1 mutant cells at 7 and 13 weeks. *P*-values were calculated by hypergeometric distribution followed by Benjamini–Hochberg correction. **d** Dot plots of the expression levels of selected genes upregulated in C1 mutant cells and matching bone-related GO terms. **e** RT-qPCR assay of the expression levels of *Sox4*, selected genes upregulated in C1 mutant cells, and osteoblastogenic markers in MSC cultures from 7- and 13-week-old *SOXC^OsxCre^* and control females before (day 0) and after 21 days in osteogenic medium. Each data point corresponds to a distinct mouse; same symbols indicate littermates. Bars and brackets represent means and standard deviations, respectively. Statistical differences were assessed by two-sided unpaired Student's *t*-tests. *P*-values lower (red) and near (black) 0.05 are indicated. **f** Matrix mineralization assay in MSC cultures that were replicates of those used in (**e**). Representative images of control and mutant cell cultures stained with Alizarin red S (dark red signals), and quantification of the percentage of stained cell culture surface (ARS^+^). Data are presented as in (**e**).

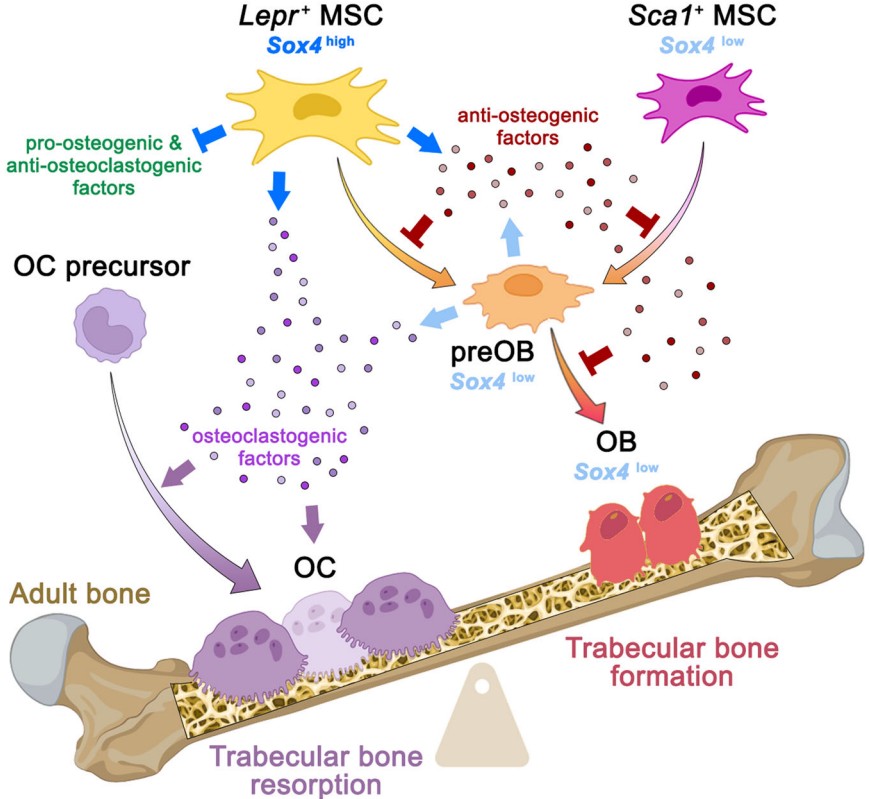

**Fig. 7 | Model for how SOX4 regulates adult trabecular bone mass.** In adult trabecular bone, SOX4 tilts the balance between bone resorption and bone formation in favor of the former. It is expressed highly in *Lepr^+^* MSCs and weakly in *Sca1^+^* MSCs, preOBs, and OBs. It most effectively modulates the expression of genes for key components of the *Lepr^+^* MSC regulatory secretome. Upregulated genes inhibit osteoblastogenesis and promote osteoclastogenesis (blue arrows), whereas downregulated genes have opposite activities (blue blunt arrows). Consequently, SOX4 creates a trabecular bone and bone marrow environment that reduces osteoblastogenesis (blunt red arrows) and favors osteoclastogenesis (thin purple arrows). MSC mesenchymal stem cell, preOB preosteoblast, OB osteoblast, OC osteoclast. Image created with BioRender.com.

Our data showed different impacts of SOXC inactivation according to bone type (vertebrae or long bones), sex, and age. In particular, *SOXC^OsxCre^* females had a more pronounced trabecular phenotype than males in long bones in mature adulthood, but both sexes had similar phenotypes in vertebrae. Ovariectomy reduced the impact of SOXC inactivation on long bone trabeculae, supporting the notion that SOX4 reduces the positive influence of female sex hormones on trabecular bone mass maintenance or vice versa that female sex hormones reduce the negative influence of SOX4. Interferon signaling may be a common target of SOX4 and sex hormones. Indeed, interferon-related genes are more downregulated in female than male mutant mice, and immune cells are more responsive to interferons in women than men[95–97]. Production of chemokines from osteoblastic cells is generally inhibited by estrogens. *Ccl2* expression is increased in ovariectomized and aged mice, and 17β-estradiol administration lessens this effect[98]. Likewise, the serum level of CCL2 is elevated in postmenopausal osteoporotic women[99]. *Cxcl9* too is upregulated in ovariectomized mice, while estrogen has the opposite effect on primary osteoblasts[86]. Thus, SOX4 and estrogen may oppose each other in modulating the expression of these genes. Similar actions can be involved in the control of other deregulated genes in SOXC-mutant cells. Both androgens and estrogens have protective effects on bone health. However, while androgens are more powerful in promoting bone formation, estrogens are more efficient in inhibiting osteoclast activity[100]. This is consistent with the facts that osteoclast numbers were low in SOXC-mutant females throughout adulthood, whereas

males showed differences only in late adulthood and resorbed trabeculae more extensively than females.

The observation that SOXC-mutant males and females have similar trabecular bone phenotypes in vertebrae but not in long bones is reminiscent of findings reported for other genes. For instance, *Sfrp1* inactivation increases the trabecular bone mass of adult mice, and this phenotype develops earlier and is more intense in female than male femurs and is equally weak in male and female vertebrae[47]. Its similarity to that of SOXC-mutant mice suggests that the strong downregulation of *Sfrp1* (and *Sfrp2*) detected in SOXC-mutant mice contributes to their phenotype. *Sfrp1* is an inhibitor of WNT signaling. Similar differences were also described in mice with other mutations altering WNT signaling[101,102]. Since WNT signaling greatly facilitates bone cell mechanotransduction[103], it was proposed that differences in mechanical loading patterns may explain differences observed between long bones and vertebrae in mice with mutations affecting WNT signaling. Males were also proposed to be less sensitive than females to variations in the WNT pathway because androgens can augment the pathway activity[104]. Based on these data, we speculate that increased WNT signaling may contribute to generating the trabecular bone phenotype of SOXC-mutant mice, and explain differences between long bones and vertebrae and between males and females.

In conclusion, this study significantly expands our understanding of the complex regulation of bone formation and resorption by revealing the SOX4 dependency of major roles of *Lepr*+ MSCs in this crosstalk. These findings should prompt further studies to decode the molecular mechanisms regulating these cells and underlying the actions and regulation of SOX4, with the ultimate goal of identifying novel therapeutic strategies for diseases affecting adult bone mass maintenance.

## Methods

### Mice

Mice were used as approved by the Cleveland Clinic and Children's Hospital of Philadelphia Institutional Animal Care and Use Committees. Mice carried *Sox4*, *Sox11*, and *Sox12* conditional null alleles[18,105] and a *Prx1Cre*[106], *OsxCre*[107] or *Col10Cre*[108] transgene or an *Acan*^CreERT2 allele[109]. Mice with *R26*^SOX4 alleles were generated as described below. Mice were genotyped by PCR of genomic DNA using KAPA HotStart Mouse Genotyping Kit (HSMGTKB, Sigma–Aldrich) and primers as listed (Supplementary Table 1). Unless otherwise stated in Results, mice featuring *OsxCre* were fed with doxycycline-containing food (200 mg doxycycline/kg food; S3888, Bio-Serv) to prevent *Cre* expression. Cre recombinase activity was induced in 21-day-old *Acan*^CreERT2/+ mice by intraperitoneal injection of tamoxifen (1 mg/10 g body weight; T5648, Sigma–Aldrich) for 4 consecutive days. Bilateral ovariectomy was performed upon mouse anesthesia by making a single midline incision on the dorsal surface, ligating the uterine horns, and removing the ovaries[110]. Success of the procedure was assessed postmortem by the absence of ovaries and reduction in uterus size.

### Generation of mice with *R26*^SOX4 allele(s)

Mice harboring a *R26*^SOX4 allele were generated by DNA homologous recombination in embryonic stem cells using a well-established strategy whereby the human SOX4 coding sequence was inserted into the *Gt(ROSA)26Sor* locus (referred to as *R26* hereafter)[111,112]. The targeting vector was built using the pBigT (21271, Addgene) and pROSA26-PA (21271, Addgene) vectors[113], and a plasmid containing a 3FLAG epitope and a linker fused in frame with the human *SOX4* coding sequence[25]. Expression of the knocked-in hSOX4 sequence was driven by a CAG hybrid promoter[114]. DNA homologous recombination in 129xBL6 F1 embryonic stem (ES) cells was performed at the Penn Vet Transgenic Mouse Core. ES cell clones harboring a proper recombination of an *R26* allele into an *R26*^SOX4 allele were identified by PCR of genomic DNA (Supplementary Table 1). Two such clones were used by the CHOP

Transgenic Core to generate mouse chimeras, from which we derived mouse lines in the C57BL/6 J genetic background. Mice were genotyped by PCR as described above.

### Radiography and microcomputed tomography

Radiographic images of the mice were acquired postmortem using a Faxitron X-ray System at 50 kV for 5 s. Length of the bones was measured on contact sides using lateral view images and built-in software. For microcomputed tomography (μCT), left femurs were fixed in 4% paraformaldehyde (PFA) in phosphate-buffered saline (PBS) for 48 h at room temperature. Most μCT analyses were performed using a SCANCO μCT 45 scanner (SCANCO Medical AG). Bones were imaged with an X-ray tube voltage of 55 kV and current of 145 μA, with a 0.5-mm aluminum filter. Voxel size was 4.5 μm and integration time was 400 ms with 2228 total slices. Cross-sectional images were reconstructed with built-in reconstruction software. Distal metaphyses were analyzed. For the trabecular bone, a volumetric region of interest (ROI) excluding cortical bone was defined starting at 100 layers (450 μm) below the lower surface of the growth plate and extending 200 layers distally (900 μm thick). For cortical bone, a slice of the diaphysis was defined starting 900 layers below the lower surface of the growth plate and extending for 100 layers distally (450 μm thick). Bones from *OsxCre* mutant mice and control littermates presented in Fig. 3a and Supplementary Fig. 7 were analyzed using an eXplore Locus SP scanner (Trifoil Imaging). Samples were imaged with an X-ray tube voltage of 80 kV, current of 80 μA, voxel size of 18 μm, exposure time of 1600 ms, and 2 × 2 detector binning. For each sample, the stage rotated through 200 degrees and collected 400 images per scan. Raw data reconstruction and analyses were performed in MicroView v. 2.2 (GE Healthcare).

### Biomechanical tests of bones

Specimens were dissected free of soft tissue and subjected to a 3-point-bending assay on a universal test frame (Instron 5543), equipped with a 50 N load cell. The supports for the test were kept at a constant length (L) of 8.80 mm throughout testing. Bones were placed onto the supports and the actuator was moved at a constant speed of 0.03 mm/s until failure. Force and displacement data were recorded throughout the test. Stiffness (slope of the linear region of the force-displacement curve, N/mm) and maximum load before failure were recorded. Flexural modulus of bones was calculated using the equation $E = FL^3/48\delta I$, where F is the maximum force, L is the span of the supports, $\delta$ is the net displacement of the actuator at maximum load, and I is the second moment of area of the cortical bone. The second moment of area was calculated using custom software that analyzed μCT images of the midshaft portions of the bones tested[115].

### Bone histomorphometry

Mice were injected intraperitoneally with calcein (20 mg/kg body weight; C0875, Sigma–Aldrich) and alizarin-3-methyliminodiacetic acid (30 mg/kg body weight; A3882, Sigma–Aldrich) nine and two days, respectively, before euthanasia. Left femurs and vertebrae were fixed in 4% PFA in PBS for 48 h at room temperature, dehydrated with acetone, and embedded in methyl methacrylate without decalcification to obtain 7-μm sections. Goldner's trichrome staining[116] was performed for visualizing bone surface, osteoid, and osteoblasts, while tartrate-resistant acid phosphatase (TRAP) staining[117] was used for osteoclast detection. Fluorescent labeling was assessed on unstained sections. Slides were imaged using ZEISS Axio Scan.Z1 scanner (Carl Zeiss). Static and dynamic histomorphometry was performed on the endosteal surfaces of the cortical femur diaphysis, and on the trabecular regions of the femur distal metaphysis and third lumbar (L3) vertebra under 20× magnification using BIOQUANT OSTEO analyzing software. Structural parameters (bone surface [BS], bone volume/tissue volume [BV/TV], trabecular diameter [Tb.Dm], trabecular number

[Tb.N], and trabecular spacing [Tb.Sp]) were obtained by averaging measurements in three consecutive sections. All parameters were calculated and expressed according to standardized nomenclature.

## Histology, RNA in situ hybridization and immunostaining

For histological analyses, right hind limbs were fixed as described above and decalcified in Morse's solution for 48 h at room temperature. Samples were dehydrated and embedded in paraffin to obtain 7-μm sections. Staining with hematoxylin and eosin was done following a standard protocol. RNA in situ hybridization was performed on deparaffinized sections using RNAscope 2.5 HD detection reagent kit-RED and a probe for mouse *Sox4* (887051, Advanced Cell Diagnostics)[118]. Slides were imaged using ZEISS Axio Scan.Z1 scanner. The blue color generated by hematoxylin counterstaining was desaturated (changed to gray) by Adobe Photoshop. For SOX4 immunostaining, sections were deparaffinized, incubated in citrate buffer pH 6.0 for 2 h at 60 °C for antigen retrieval, and treated with M.O.M.® (Mouse on Mouse) Immunodetection Kit (BMK-2202, Vector Biolaboratories). Protein was detected using a monoclonal anti-SOX4 antibody (dilution 1:500; MA5-31423, Thermo Fisher Scientific) and a biotinylated anti-mouse IgG2b antibody (dilution 1:3000; ab97248, Abcam). Signals were amplified with a TSA-Fluorescein kit (NEL741001KT, Akoya Biosciences), and slides were counterstained with DAPI and mounted using ProLong™ Gold Antifade Mountant (P36930, Thermo Fisher Scientific). Images were acquired using a Leica TCS SP8 confocal microscope.

## Serum assays

After mouse euthanasia with $CO_2$ asphyxiation, blood was collected by heart puncture using a 1.0-mL syringe and collected in microfuge tubes. Tubes were incubated at room temperature for 1 h before centrifuging at 3000 rpm for 10 min at 4 °C. Supernatants (containing sera) were aliquoted and stored at −80 °C. Serum levels of RANKL (MTR00; R&D Systems), OPG (MOP00; R&D Systems), P1NP (NBP2-76466, R&D Systems) and CTX-1 (NBP2-69074, R&D Systems) were measured according to manufacturer's instructions.

## Bone and bone marrow cell isolation

Femurs and tibiae were collected from 7- and 13-week-old *SOXC*$^{OsxCre}$ and control female littermates. Muscle and periosteum tissues were removed by thoroughly scraping the bone surfaces with a scalpel. Epiphyses were removed and bone marrow was flushed out using a syringe with 1% BSA in PBS. Bone shafts were sliced in halves longitudinally and further washed with the same solution. Tissues were then cut in ~1-mm³ fragments and digested with four alternate incubations in Liberase™ (1 Wunsch unit/mL in Hank's Balanced Salt Solution; Fisher Scientific) and 5 mM EDTA in calcium- and magnesium-free PBS with 0.1% BSA for 20 min at 37 °C each. Cell suspensions were then passed through a 70-μm strainer and sorted to enrich the mesenchymal populations. Briefly, cells were incubated with biotinylated antibodies against hematopoietic markers, including the mouse lineage cell depletion kit (130-090-858, Miltenyi Biotec), and anti-TER-119 (13-5921-85, Thermo Fisher Scientific), anti-CD71 (130-109-572, Miltenyi Biotec), and anti-CD3e (130-093-179, Miltenyi Biotec) antibodies. Cells were then incubated with anti-biotin magnetic microbeads (130-090-485, Miltenyi Biotec) and sorted through LS columns (130-042-401, Miltenyi Biotec) containing ferromagnetic spheres and mounted on a MACS separator (130-090-976, Miltenyi Biotec). An additional round of sorting was performed using anti-CD45 microbeads (130-052-301, Miltenyi Biotec). Non-labeled cells were eluted and resuspended in PBS with 0.5% BSA before processing for scRNA-seq assays.

## Single-cell RNA-seq assays

Hematopoietic cell-depleted bone and bone marrow cell suspensions were counted with a hemocytometer and encapsulated into emulsion droplets using 10× Genomics Chromium Controller (10× Genomics). Libraries were constructed for scRNA-seq using Chromium Single Cell 3′ v3 Reagent Kit (10× Genomics) according to the manufacturer's protocol and were sequenced on an Illumina NovaSeq sequencer. Data were processed using the 10× Genomics workflow. In brief, Cell Ranger v6.1.2 (10× Genomics) was used for demultiplexing, barcode assignment, and unique molecular identifier (UMI) quantification. Downstream analyses were performed using Seurat v4[119]. Cells with >6000 expressed genes, <200 expressed genes, and >10% mitochondrial transcripts were excluded. Doublets were removed using DoubletFinder[120]. Data were normalized and integrated. Principal Component Analysis (PCA) and Uniform Manifold Approximation and Projection (UMAP) were performed. Marker genes for cluster identification and differentially expressed genes among control and mutant cells were determined using the "FindMarkers" function, applying a logarithmic fold change threshold ≥0.25 and *P*-value ≤ 0.05. Gene Ontology (GO) enrichment analyses for biological processes were performed on lists of differentially expressed genes using enrichGO function in clusterProfiler[121]. Output data were considered statistically significant for *p*-values ≤ 0.05 after false discovery rate correction using the Benjamini−Hochberg procedure. Gene lists obtained from GO analysis were used to assess module scores in control and mutant clusters using "AddModuleScore" function. Statistical analysis for module scores was performed using the Mann−Whitney test. Gene regulatory network (GRN) analysis was performed using SCENIC[44]. Only regulons including more than 20 genes and with high-confidence annotations were retained. Extended analysis of the GRN for interferon-dependent genes was performed with iRegulon in Cytoscape[69]. Trajectory and pseudotime analyses were performed with Slingshot[122] following the condiments workflow (available at https://hectorrdb.github.io/condimentsPaper/index.html), and assigning the osteoblast cluster (C6) as the trajectory endpoint. Statistical differences in trajectories were assessed using the "progressionTest" function. For RNA velocity analysis, counts and moments for spliced and unspliced reads were generated with velocyto[123] and scVelo[124], respectively. Estimation of RNA velocity was then performed with cellDancer[125] using a gene list for osteoblast differentiation (GO term: 0001649) and following instructions for transcriptional boost analysis. Previously generated data[35] were downloaded from NCBI Gene Expression Omnibus (accession number GSE145477) and analyzed with the Seurat package.

## Colony-forming unit and osteogenic differentiation assays

Bone marrow was flushed out from the mouse femurs and tibiae using DMEM with 10% fetal bovine serum (FBS, F2442, Sigma−Aldrich) (complete medium). It was centrifuged at 300 g for 5 min at room temperature and the pellet was resuspended in complete medium. For colony-forming unit (CFU-F) assays, the cells from all four bones were plated in a T25 flask and grown for 10 days. Cells were then fixed with 4% PFA in PBS and stained with 1% methyl violet (LabChem™). The numbers and sizes of colonies were assessed using ImageJ software. For osteogenic differentiation, cells were plated in 12-well dishes and grown in complete medium for 10 days. Cells were then cultured in osteogenic medium (αMEM with 10% FBS, 50 μg/mL ascorbic acid, 5 mM β-glycerophosphate, and 100 mM dexamethasone) for 21 days. To visualize the amount of mineralized matrix, cells were fixed in 4% PFA for 10 min, stained with 1% Alizarin Red S, pH 4.2, for 30 min, and washed with water. The percentage of Alizarin red-stained area was measured using ImageJ software.

## Cell treatment with interferon-β

C3H/10T1/2 cells (CCL-226, ATCC, Manassas, VA, USA) were cultured in complete medium (DMEM with 10% FBS). The day of the experiment, 120,000 cells were plated in 12-well dishes and transfected with 30 nM *Sox4* siRNA (assay s74178, Thermo Fisher Scientific) or negative control

siRNA (AM4620, Thermo Fisher Scientific) in Lipofectamine RNAiMAX (13778-075, Thermo Fisher Scientific) according to the manufacturer's instructions. After 8 h, cells were treated with mouse IFN-β (8234MB010, R&D Systems) at indicated doses. They were analyzed 20 h later.

## Quantitative reverse transcription PCR (RT-qPCR)

RT-qPCR assays were performed on cells directly after sorting from bones or following culture in vitro. Cells were lysed in TRIzol™ reagent (15596026, Thermo Fisher Scientific) and RNA was extracted using RNeasy Plus Micro Kit (74034, Qiagen, Hilden, Germany). RNA was quantified using NanoDrop 2000 (Thermo Fisher Scientific), and 100–500 ng of RNA was used for reverse transcription with the High-Capacity cDNA Reverse Transcription Kit (4368814, Thermo Fisher Scientific). 1 μL of cDNA was used for assessing gene expression using PowerUp™ SYBR™ Green Master Mix (A25742, Thermo Fisher Scientific) and primers listed in Supplementary Table 2.

## Western blot assay

Protein extracts from long bones were prepared in RIPA buffer (150 mM NaCl, 1% IGEPAL CA-630, 1% SDS, 25 mM Tris-HCl pH 7.4) supplemented with Halt™ Protease and Phosphatase Inhibitor Cocktail (78440, Thermo Fisher Scientific) using a Precellys homogenizer (Bertin Technologies). Protein extracts from cell cultures were prepared in RIPA buffer by manual homogenization. Total protein content was quantified using the DC protein assay kit (5000111, Bio-Rad laboratories). 20 μg of protein were electrophoresed in 10% polyacrylamide gels with sodium dodecyl sulfate (SDS) and transferred to iBlot2 PVDF membranes using iBlot Dry Blotting System (Thermo Fisher Scientific). Blots were blocked in 5% non-fat dry milk in TBST buffer (20 mM Tris-HCl pH 7.6, 137 mM NaCl, 0.1% Tween-20) and then incubated in the same buffer with antibodies against SOX4 (dilution 1:2000; MA5-31423, Thermo Fisher Scientific) or β-actin (dilution 1:2000; SC47778, Santa Cruz Biotechnology) at 4 °C overnight. The next day, blots were incubated with peroxidase-conjugated anti-mouse antibody (dilution 1:5000; 1706516, Bio-Rad Laboratories) in TBST buffer for 1 h at room temperature. For the detection of FLAG epitope, blots were incubated with a peroxidase-conjugated anti-FLAG antibody (dilution 1:10000; A8592, Sigma–Aldrich) for 1 h at room temperature. Peroxidase signals were generated using Clarity Western ECL Substrate (Bio-Rad Laboratories) and were visualized in a ChemiDoc Gel Imaging System (Bio-Rad Laboratories).

## Statistical analysis

Analysis of scRNA-seq data was performed using Seurat built-in statistic tests as described above. For all the other data, statistical analyses were performed using GraphPad Prism software. Results are depicted as mean ± SD, if not stated otherwise. Experimental group sizes ($n$) are reported and defined in the Source Data file. Two-tailed Student's $t$-tests were used for two-group comparisons, while one-way or two-way ANOVA was used for multi-group comparisons. Mann–Whitney tests were used to assess differences in regulon activity scores and module scores for specific gene sets between control and mutant cell populations. $P$-values ≤ 0.05 were considered significant ($*p \leq 0.05$; $**p \leq 0.01$; $***p \leq 0.001$; and $****p \leq 0.0001$).

## Reporting summary

Further information on research design is available in the Nature Portfolio Reporting Summary linked to this article.

## Data availability

The scRNA-seq data generated in this study are available at the NCBI Gene Expression Omnibus repository (GSE241637). Source data are provided with this paper.

## Code availability

scRNA-seq analyses were performed using standard pipelines (specific code scripts are available upon request).

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

## Acknowledgements
This work was funded by the NIH/NIAMS R01AR68308 grant to V.L. We thank K. Duncan for bioinformatics assistance, C. de Charleroy for technical assistance, and C. O'Brien, M. Almeida, L. Qin, and E. Schipani for scientific and technical advice. We also thank K. Jepsen and the University of Michigan MicroCT Core for initial microCT analysis of SOXC-mutant bones (NIH/NIAMS P30AR069620 grant); R. Mauck and the PCMD Biomechanics Core for mouse bone biomechanical testing (NIH/NIAMS P30AR069619 grant); C. Lengner, S. Adams and the Penn Vet transgenic mouse core for generating $R26^{SOX4/+}$ ES cell clones; A. Harman and the CHOP transgenic core for generating $R26^{SOX4/+}$ mouse chimeras; and the CHOP CAG core for processing samples for scRNA-seq.

## Author contributions
V.L. conceived the project. V.L. and M.A. supervised the project, designed the experiments, and wrote the manuscript. M.A. and A.K. equally contributed to the experiments. All authors analyzed the data and reviewed and approved the manuscript.

## Competing interests
The authors declare no competing interests.
