## [Peer Review File · Nature Communications]

SOXC are critical regulators of adult bone massReviewer #1 (Remarks to the Author):

In this study, Angelozzi et al. studied the role of SOXC family TFs, including SOX4, SOX11 and SOX14, in bone mass regulation by loss- and gain-of-function analyses. They found that SOXC increases cortical bone width, thickness and strength in both sexes, while SOX4 reduces trabecular bone mass in females. SOX4 favors bone resorption over bone formation by decreasing osteoblastogenesis and increasing osteoclastogenesis. They also performed scRNA-seq to explore the underlying molecular mechanisms, and proposed that upregulation of type I interferon stimulated genes by SOX4 could explain the trabecular bone phenotypes described above.

Overall, the authors performed fairly comprehensive analyses of the bone phenotypes, which provide convincing evidence to demonstrate the critical role of SOXC in bone mass maintenance. However, several concerns need to be addressed before accepting the manuscript for publication.

Major points

1. Fig S2 is exactly the same as Fig 2, which is a big mistake. Without seeing the bona fide Fig S2, the conclusions cannot be justified.
2. Whereas they found increased trabecular bone mass in female femurs only, the vertebrate trabecular bone mass was increased in both sexes (Fig. S3). How do they explain that sexual dimorphism only occurred in appendicular bones?
3. Characterization of the cortical bone phenotypes are incomplete. Static and dynamic histomorphometric analyses on cortical bones are equally important. Whereas a higher ratio of P1NP/CTX-1 and lower ratio of RANKL/OPG is consistent with increased trabecular bones, it is contradictory to decreased cortical bones in the absence of SOXC. Since these serum markers are systemic factors, how do they reconcile this discrepancy?
4. The authors performed scRNA-seq by combining bone fragments and bone marrows. Therefore, the 17 cell clusters should include periosteal, endosteal and bone marrow cells (Fig. 5B). A recent study by Mo et al. identified a Sca1+ periosteal population that highly express Col3a1 (EMBO J, 2022, Fig. 1 and Fig. S1), which is very similar to cluster 1 (Sca1+ BMMSC) in this study (Fig. 5B and C, highly expressing Col3a1 and Periostin). If that is the case, up-regulation of ECM organization and ossification genes in cluster 1 after SOXC deletion would contrast with decreased cortical bones, which is maintained by periosteal and endosteal osteoprogenitors.
5. Since RNA velocity is a pseudotime method that relies on calculating different mRNA isoforms, it is not ideal or accurate enough to infer lineage relationships for 10X Genomics-based scRNA-seq that only reads 3' end but not full-length mRNAs. More reliable pseudotime methods such as Monocle or Slingshot are suggested. Accordingly, it is rather premature to conclude that Lepr-high BMMSCs and preOBs are derived from Sp7-low/Lepr-low progenitors (line 355 and Fig. 8).

Minor points:

1. Please avoid drawing strong claims on SOX4 (eg. master regulator, line 365), as it is incomparable to master osteogenic TFs such as Runx2 or Sp7.
2. The font size in most figures is too small to be seen (eg. Fig. 6B and H). Please adjust for better readability.
3. In Fig. 4E, co-staining of Lepr antibody could help demonstrate SOX4 expression in Lepr+ BMMSCs.
4. Doublet exclusion, cell-cell interaction and regulon analyses are recommended to further improve the scRNA-seq part.

Reviewer #2 (Remarks to the Author):

This paper focuses on the analysis of Sox proteins as regulators of bone mass. To set the stage, bone mass is the outcome of two opposite functions in bone, bone formation and bone resorption. Each of these two functions is regulated in a cell-autonomous and non-cell-autonomous manner during development and postnatally. Among the transcription factors that affect bone formation by osteoblasts or bone resorption by osteoclasts, one should at the very least cite Runx2, Sp7, AP-1, and Pu-1, etc...as their respective inhibition profoundly affects osteoblasts and/or osteoclast differentiation and not ignore them to remain focused on the Sox genes. Among the non-cell-autonomous factors affecting bone mass, one should cite PTH, estrogens, leptin, innervation and

many others.

In this paper, the roles of the transcription factors of the Sox family and, in particular, of the Sox_c of Sox proteins during cell differentiation in cells and in the skeleton in particular are examined. In short, mice lacking all Sox_c proteins present a clear but modest decrease in bone mass affecting cortical and trabecular bones, axial and peripheral skeleton. These mice do not present fracture but have abnormal osteoblast and osteoclast numbers. There is a mild dichotomy in the severity of the phenotype between males and females. A transcriptional analysis proposes a mechanistic explanation for the phenotypes presented. The quality of the data presented is outstanding, and the quality seems a priori-intimidating. What is more important is the interpretation of these data sometimes to be modified.

To put this contribution in perspective, one has to mention the previous work that looked at the role of Sox_c proteins in the skeleton. To be fair, the contributions of these previous studies, even if cited in the manuscript, are somewhat minimized or, for Sox₄, misrepresented. Indeed, according to reference 24, Sox₄ haploinsufficiency does not affect Runx2 but only Sp7. For somebody outside the field, this handling of the previous literature does not appear justified, especially since, in one case, one studied the Sox₄ role in isolation (reference 24), whereas in the other case, one looks at all Sox_c proteins. This contrasts with how the authors justify some of their claims by citing abstract (reference 37) or papers published in journals rarely cited.

Overall, and this applies to all figures, there are too many panels per figure, some with positive others with negative data. If anything, this removes some weight to the quality of the paper and some of these data could easily find their home in supplemental figures. This way of presenting is indeed overwhelming but is also less convincing than it could be anything it hurts the paper. Do we really need to see cortical thickness represented in two different ways in Figure 1, for instance? This artificially takes a lot of space and leaves unaddressed the more important question of why the mutant mice die perinatally. Given the nature of the Cre used in this study, this perinatal lethality, and low body weight at weaning, one is surprised by the absence of any histological or molecular analysis during development or in the first weeks of life.

If we take a step back there is no need to have one figure for cortical bone, one figure for trabecular bone analyzed by microCT, and another one for trabecular bone analyzed by classical histological methods. There is instead a need for a more synthetic presentation. This is even more important since the author failed to identify a mechanism for the more severe phenotype in female than in male mice. In contrast, the demonstration that among the Sox_c protein, only Sox₄ regulates trabecular bone mass is convincingly done in a single figure, as it should.

The last part of this study includes a transcriptomic analysis performed at 7 or 13 weeks of age in bone marrow mesenchymal stem cells of control and mice lacking all Sox_c proteins. In view of the bone phenotype of the mutant mice, increased bone formation, and decreased bone resorption this analysis does not provide clear answers to explain why this dual histological phenotype take place. Here again, figures are cluttered with negative results that add little to the story. Why is bone formation decreased in the mutant mice? Does Sox₄ regulate Type II collagen gene expression in a more direct measure than the one presented? Do Sox_c or does Sox₄ alone regulate the expression of the interferon genes in osteoblasts in culture? How is the expression of Ank and alkaline phosphatase in the mutant bone and osteoblasts? What does the author mean by "Sox_c proteins generate a bone milieu delaying osteoblast differentiation but favoring bone mineralization? Rather, it would have been more important to perform in vivo experiments supporting a role of interferon in the ontogeny of this phenotype.

Point-by-point response to the Reviewers

We are very thankful to the reviewers for their time and expertise in assessing our manuscript and for their constructive recommendations to improve it. We have now addressed each one of their concerns and have followed almost all recommendations in a revised manuscript, as explained in our point-by-point response below. We believe that our study is now much stronger and easier to understand, and we hope that the reviewers and editors will agree.

Major changes include:

- We have streamlined and reorganized the figures as recommended by the reviewers. We have now grouped in the same figures all analyses used to study and describe specific phenotypes. For instance, all the μ CT and histomorphometry analyses of cortical bones are in Figure 1, and an analogous organization has been adopted for trabecular bones in Figure 2. Similarly, molecular data for gene expression changes in *Lepr*⁺ MSCs are now presented in one figure, Figure 5. To make this possible, we kept the results essential to convey the key messages of the study in the main figures, and moved auxiliary data to supplementary figures. We now have 7 main figures instead of 8, and 11 supplemental figures instead of 8.
- We have reduced the emphasis on the sex dichotomy for the trabecular bone phenotype and on the involvement of SOX4 in interferon signaling since future studies are needed to fully elucidate these processes. This was achieved by making text changes, moving data from ovariectomized mice to a supplemental figure (Figure S6), and including the data on the interferon pathway in the same figure as those on other pathway changes in *Lepr*⁺ MSCs (Figure 5).
- We have added data from static and dynamic histomorphometry analyses of the endosteal surfaces of cortical bones. These data complement the μ CT analyses and help explain the cortical bone phenotype of SOXC mutant mice.
- We have complemented data from Gene Ontology analysis with data from gene regulatory network analysis by SCENIC. Together, these analyses give new insight into possible direct and indirect targets of SOX4 and into the etiology of the trabecular phenotype of SOXC mutant mice.
- We have added trajectory analyses by Slingshot. Together with data from RNA velocity analysis, the new data further support a role for SOXC in delaying osteoblast differentiation of *Lepr*⁺ and *Sca1*⁺ MSCs.
- We have made many changes throughout the manuscript to address all reviewers' concerns and to ensure that our study is easier to read and understand. These changes are so numerous that we did not highlight them in the manuscript file.

Reviewer #1 (Remarks to the Author):

In this study, Angelozzi et al. studied the role of SOXC family TFs, including SOX4, SOX11 and SOX14, in bone mass regulation by loss- and gain-of-function analyses. They found that SOXC increases cortical bone width, thickness and strength in both sexes, while SOX4 reduces trabecular bone mass in females. SOX4 favors bone resorption over bone formation by decreasing osteoblastogenesis and increasing osteoclastogenesis. They also performed scRNA-seq to explore the underlying molecular mechanisms, and proposed that upregulation of type I interferon stimulated genes by SOX4 could explain the trabecular bone phenotypes described above.

Overall, the authors performed fairly comprehensive analyses of the bone phenotypes, which provide convincing evidence to demonstrate the critical role of SOXC in bone mass maintenance. However, several concerns need to be addressed before accepting the manuscript for publication.

Major points:

1. Fig S2 is exactly the same as Fig 2, which is a big mistake. Without seeing the bona fide Fig S2, the conclusions cannot be justified.

Thank you for noticing this mix-up, which inadvertently happened during the uploading of the figures. We have now uploaded the figure containing the correct data. Please note that the original Fig. S2 is now Fig. S3 in the new version of the manuscript. We hope that the reviewer will agree with the conclusions that we reached regarding these data.

2. Whereas they found increased trabecular bone mass in female femurs only, the vertebrate trabecular bone mass was increased in both sexes (Fig. S3). How do they explain that sexual dimorphism only occurred in appendicular bones?

We thank the reviewer for pointing out the differential effect of SOXC inactivation on trabecular bone in femurs versus lumbar vertebrae and in male versus female mice. We are sorry for not addressing these intriguing phenotype differences in the original version of the manuscript. Such sex-dependent and site-dependent phenotypes have been reported in other studies. For instance, *Sfrp1* inactivation resulted in increased trabecular bone mass in adult mice, and this phenotype developed earlier and was more intense in the femurs of females than males and was equally weak in the vertebrae of males and females (Bodine et al., Mol Endocrinol, 2004). This phenotype is similar to that observed in SOXC mutant mice, suggesting that the strong downregulation of *Sfrp1* (and *Sfrp2*) detected in SOXC mutant mice contributed to their phenotype. *Sfrp1* is an inhibitor of WNT signaling. Similar differences were also described for other mice with mutations altering WNT signaling (Noh et al., PLoS ONE, 2009; Albiol et al., Calc Tissue Int, 2020). Since WNT signaling greatly facilitates bone cell mechanotransduction (Choi et al., Bone, 2021), it was proposed that differences in mechanical loading patterns may explain differences observed between long bones and vertebrae in mice with mutations affecting WNT signaling. It was also proposed that males are less sensitive than females to variations in the WNT pathway because androgens can augment the pathway activity (Liu et al., Ann. N.Y. Acad. Sci., 2007). Based on these findings, we speculate that increased WNT signaling is at least one mechanism underlying the trabecular bone phenotype of SOXC mutant mice, including differences in long bones versus vertebrae and in males versus females. We have now added a paragraph in the Discussion (second to last) to address this point. Moreover, we have revised the Results section entitled "SOXC inhibit adult trabecular bone mass accrual" to more accurately report these findings. This includes the following sentence: Histomorphometry of lumbar vertebrae showed an increase in BV/TV and trabecular number in 13- and 52-week-old SOXC^{OsxCre} mice, but differently from long bones, this increase was similar in males and females (Fig. S5a,b). Males also showed a larger trabecular diameter".

3. Characterization of the cortical bone phenotypes is incomplete. Static and dynamic histomorphometric analyses of cortical bones are equally important. Whereas a higher ratio of P1NP/CTX-1 and lower ratio of RANKL/OPG is consistent with increased trabecular bones, it is contradictory to decreased cortical bones in the absence of SOXC. Since these serum markers are systemic factors, how do they reconcile this discrepancy?

We now present both static and dynamic histomorphometry data for the cortex endosteal surfaces at regions equivalent to those used for μ CT analysis. The new data show that despite having the same number of osteoblasts, the mineralization and bone formation rates in mutant bones are significantly reduced at 5 weeks with respect to controls in both sexes, likely explaining the early appearance of the cortical bone phenotype. We added these data in Fig. 1c-e.

Regarding the apparent discrepancy between the cortical bone phenotype and the P1NP/CTX-1 and RANKL/OPG ratios, it is important to note that SOXC mutants fail to enlarge their cortical bones as much as control mice between 3 and 5 weeks of age, thus before we measured the serum P1NP/CTX-1 and RANKL/OPG ratios, and that they maintained their acquired defect through adulthood, but did not worsen it. Our analysis thus lacked the age at which mutant cortical cells are impaired. This likely explains the apparent contradiction between the cortical bone mass phenotype and the ratios of serum markers. Additionally, since the adult turnover of trabecular bone is more intense than that of cortical bone due to a proportionately larger surface area (Seeman. *J Gerontol A Biol Sci Med Sci.* 2013; Tower et al., *Bone* 2015), it is reasonable to speculate that the trabecular bone phenotype weighs more on the levels of P1NP, CTX-1, RANKL and OPG than the cortical bone phenotype. We have modified the Results section and added a sentence (The cortical bone phenotype of mutant mice unlikely affected levels of serum markers significantly since this phenotype was stable by the time (5 weeks) of the analysis and since the adult turnover of trabecular bone is more intense than that of cortical bone^{33,34}) to address this point.

4. The authors performed scRNA-seq by combining bone fragments and bone marrows. Therefore, the 17 cell clusters should include periosteal, endosteal and bone marrow cells (Fig. 5B). A recent study by Mo et al. identified a *Sca1*⁺ periosteal population that highly express *Col3a1* (EMBO J, 2022, Fig. 1 and Fig. S1), which is very similar to cluster 1 (*Sca1*⁺ BMMSC) in this study (Fig. 5B and C, highly expressing *Col3a1* and *Periostin*). If that is the case, up-regulation of ECM organization and ossification genes in cluster 1 after SOXC deletion would contrast with decreased cortical bones, which is maintained by periosteal and endosteal osteoprogenitors.

While *Sca1*⁺ (*Ly6a*) cells described by Mo et al. have a transcriptome similar to that of *Sca1*⁺ cells in our dataset, we believe that our cells are of bone marrow rather than periosteal origin for the following reasons:

1) As written in Materials and Methods, “Muscle and periosteum tissues were removed by thoroughly scraping the bone surfaces with a scalpel” before proceeding with tissue digestions. Thus, it is unlikely that periosteal cells constitute a significant population in our scRNA-seq dataset. We have also added this information in the Results section: “We collected femurs and tibiae from two pairs of 7- and 13-week-old SOXC^{OsxCre} and control sisters, thoroughly scraped the periosteum away, enzymatically digested the cortical bone, trabecular bone, and endosteal bone marrow tissues, and immunomagnetically sorted out hematopoietic cells (Fig. 4a)”.

2) SCA1⁺ mesenchymal cells have been recognized in mouse bone marrow for a long time. Mostly known as PαS (PDGFRα⁺SCA1⁺) cells, they are the main contributors of cell colony formation in vitro and possess a tri-lineage differentiation potential (Morikawa et al., *J Exp Med.* 2009). These features are in line with our in vitro experiments using cells that are unlikely to be contaminated by periosteal cells because they were isolated by flushing bone marrow out. Cells in these cultures expressed high levels of *Ly6a* (encoding SCA1), *Col3a1* and *Postn*, in line with scRNA-seq data.

3) Bone marrow-derived SCA1⁺ cells exhibiting transcriptome profiles similar to those of *Ly6a*⁺ cells in our datasets were reported in other studies, as for instance in Zhong et al., *Elife* 2020 (where the cells were called EMPs, early mesenchymal progenitors) and in Helbling et al., *Cell Rep* 2019 (where the cells were called PαS cells). We have now added to the description of our C1 cluster of SCA1⁺ cells that these cells are “also known as PDGFRA1⁺SCA1⁺ or PαS MSCs (Morikawa et al., 2009) and as early mesenchymal progenitors or EMPs (Zhong et al., 2020)”.

5. Since RNA velocity is a pseudotime method that relies on calculating different mRNA isoforms, it is not ideal or accurate enough to infer lineage relationships for 10X Genomics-based scRNA-

seq that only reads 3' end but not full-length mRNAs. More reliable pseudotime methods such as Monocle or Slingshot are suggested. Accordingly, it is rather premature to conclude that *Lepr*-high BMMSCs and preOBs are derived from *Sp7*-low/*Lepr*-low progenitors (line 355 and Fig. 8).

We agree that RNA velocity may not be the most suitable method to infer trajectories with 10X Genomics 3' tag sequencing data. However, it gives information that other tools do not provide, such as the directionality of the trajectories and the kinetics at which cell transitions occur (Bergen et al., *Mol Syst Biol.* 2021). This is important in our study since SOXC mutant cells reach their final state faster, but do not take different or new trajectories with respect to control cells. We would like to emphasize that we performed RNA velocity analysis using a recently published package called "CellDancer" (Li et al., *Nat Biotechnol.* 2023), which implements the "velocity" (La Manno et al., *Nature.* 2018) and "scVelo" (Bergen et al., *Nat Biotechnol.* 2020) methods to most accurately estimate velocities in transcriptional boost circumstances, as it is the case for osteogenic differentiation. Nevertheless, we acknowledge the fact that RNA velocity has weaknesses and, as for other pseudotime methods, it remains a predictive tool rather than a conclusive method to assess cell trajectories. For these reasons, we have now performed Slingshot trajectory analyses and present them in panel a of a new Fig. 6, while RNA velocity data are shown in panel b. The data from the two types of analyses support each other in predicting that SOXC inactivation does not change the cell trajectories from *Lepr*⁺ and *Sca1*⁺ MSCs to osteoblasts, but significantly accelerate them. Panels c to f complement these data and those shown in Fig. 5 for *Lepr*⁺ MSCs and osteoprogenitors. They show namely that GO analysis of upregulated genes in the mutant C1 cell populations primarily yielded bone formation, osteoblast differentiation and related pathways, and that SOXC mutant bone marrow-derived MSCs, which are primarily *Sca1*⁺, upregulated genes associated with osteoblast differentiation.

Minor points:

1. Please avoid drawing strong claims on SOX4 (eg. master regulator, line 365), as it is incomparable to master osteogenic TFs such as *Runx2* or *Sp7*.

We no longer refer to SOXC and SOX4 as master regulators. We use such words as "key players".

2. The font size in most figures is too small to be seen (eg. Fig. 6B and H). Please adjust for better readability.

We have now increased the font size of text in figures as much as possible and necessary to facilitate reading, even in prints.

3. In Fig. 4E, co-staining of *Lepr* antibody could help demonstrate SOX4 expression in *Lepr*⁺ BMMSCs.

As suggested by the reviewer we performed LEPR and SOX4 co-immunostaining on bone sections from 7-week-old female mice. As shown in representative pictures shown below, LEPR staining (red) in the bone marrow area was consistent with previously published data from the Morrison group (Zhou et al., *Cell Stem Cell.* 2014), and SOX4 staining (white) was found, as expected, in terminal hypertrophic chondrocytes, bone-lining osteoblasts, bone-embedded osteocytes, and a subset of marrow cells. In addition to a few marrow cells (orange arrows) positive both for SOX4 (in cell nuclei) and for LEPR (in cell membranes), we also saw many cells that were positive only for SOX4 or only for LEPR. This is consistent with the facts that LEPR⁺ MSCs account for only ~0.3% of marrow cells (Zhou et al., *Cell Stem Cell.* 2014) and that SOX4 and LEPR are expressed in different marrow cell types. Indeed, as we reported in the main text,

SOX4 is expressed by hematopoietic progenitors, B-cells, and endothelial cells, and LEPR is expressed in a subset of hematopoietic stem cells and in monocytes, macrophages, and leukocytes (La Cava and Matarese, Nat Rev Immunol. 2004; Trinh et al., Leukemia. 2021). So, co-staining for these two proteins would be a suitable method to show co-expression of LEPR and SOX4 in bone marrow MSCs only if markers restricted to MSCs could also be stained for. Reaching this goal could be challenging. Thus, we believe that scRNA-seq is the best approach to show co-expression of *Sox4* and *LepR* in bone marrow MSCs. Since our transcriptomic findings on *LepR/Sox4* co-expression in these cells reproduced published findings, we did not include these co-staining data in the manuscript. We hope that the reviewer will agree with our decision.

Figure legend. (a) Immunostaining of bone sections from 7-week-old female mice for LEPR (red signal) and SOX4 (white signal). Cell nuclei are stained with DAPI (blue signal). (a) Top left, border (dotted line) between the growth plate (GP) and primary ossification center (POC); top middle, trabecular bone (TB) and bone marrow (BM); and top right, cortical bone (CB) and periosteum (P). Bottom images (a' and b') are high magnifications of the regions boxed in the top middle image. Arrowheads point to specific cell types, as indicated. (b) Images published in Figure 1 by Zhou et al. (Cell Stem Cell, 2014). Labeling was added to the images to locate the different tissues (GP, POC, BM, TB and CB).

4. Doublet exclusion, cell-cell interaction and regulon analyses are recommended to further improve the scRNA-seq part.

We thank the reviewer for these excellent suggestions. We have now followed them as follows.

We removed putative doublets using the “DoubletFinder” package (McGinnis et al., Cell Syst. 2019). This step removed an average of 625 ± 62 cells per sample and across all clusters. This change did not significantly affect the data and our conclusions regarding the effect of SOXC inactivation on the transcriptome of bone and bone marrow cells.

We have performed Regulon analyses (SCENIC) for the control and mutant clusters most affected by SOXC inactivation. These data are presented in the new Figure 5 and Table S4. In summary, a SOX4 regulon was identified as being prominent in *Lep^{r+}* MSCs and to include several of the most downregulated genes in mutant cells. This suggests that these genes could be direct targets of SOX4. Other prominent regulons in these cells that were affected in mutant cells were found for transcription factors with important roles in bone formation and interferon-related pathways. Extended analysis with iRegulon suggests that SOX4 may modulate the expression of IFN-dependent genes through promoting the expression of STAT1 and JAK2. These findings matched the pathways identified through GO analyses. They help predict direct and indirect links between SOX4 and bone mass regulatory networks. We thus believe that they add substantial value to our study.

Reviewer #2 (Remarks to the Author):

This paper focuses on the analysis of Sox proteins as regulators of bone mass. To set the stage, bone mass is the outcome of two opposite functions in bone, bone formation and bone resorption. Each of these two functions is regulated in a cell-autonomous and non-cell-autonomous manner during development and postnatally. Among the transcription factors that affect bone formation by osteoblasts or bone resorption by osteoclasts, one should at the very least cite Runx2, Sp7, AP-1, and Pu-1, etc...as their respective inhibition profoundly affects osteoblasts and/or osteoclast differentiation and not ignore them to remain focused on the Sox genes. Among the non-cell-autonomous factors affecting bone mass, one should cite PTH, estrogens, leptin, innervation and many others.

We have added several sentences in the Introduction to address this reviewer’s recommendation while trying to keep the manuscript focused. Of note, many key factors involved in the control of adult bone turnover are described in the Results and Discussion as their expression is or is not affected by SOXC inactivation.

In this paper, the roles of the transcription factors of the Sox family and, in particular, of the Sox proteins during cell differentiation in cells and in the skeleton in particular are examined. In short, mice lacking all Sox proteins present a clear but modest decrease in bone mass affecting cortical and trabecular bones, axial and peripheral skeleton. These mice do not present fracture but have abnormal osteoblast and osteoclast numbers. There is a mild dichotomy in the severity of the phenotype between males and females. A transcriptional analysis proposes a mechanistic explanation for the phenotypes presented. The quality of the data presented is outstanding, and the quality seems a priori-intimidating. What is more important is the interpretation of these data sometimes to be modified.

We thank the reviewer for appreciating the quality of our work. We believe that he/she meant to write that the quantity, rather than quality, of data seemed a priori-intimidating. It is indeed undeniable that our study involved many mice of both sexes and at different ages, and that we performed assays at tissue, cell and molecular levels for many of them. These extensive analyses were justified by the fact that the importance of the three SOXC genes in bones beyond

developmental formation had not been studied in depth to this date and that bone formation and turnover are dynamic processes whose features and regulation are context-dependent, namely varying according to sex, age and anatomical location. We believe that our extensive analyses are rewarding as they have identified differential roles for SOXC in different contexts. For instance, while SOXC mutant mice are mildly impaired in enlarging their cortical bones during juvenile development, females markedly increase their long bone trabecular bone mass throughout adulthood, males only transiently increase their long bone trabecular bone mass in early adulthood, and both male and female adults show increased trabecular bone mass in vertebrae. While our study answered a number of questions, it also opened new fields of investigations by providing new mouse models and raising many new questions. We hope that it will inspire many new and important studies.

To put this contribution in perspective, one has to mention the previous work that looked at the role of Sox proteins in the skeleton. To be fair, the contributions of these previous studies, even if cited in the manuscript, are somewhat minimized or, for Sox4, misrepresented. Indeed, according to reference 24, Sox4 haploinsufficiency does not affect Runx2 but only Sp7. For somebody outside the field, this handling of the previous literature does not appear justified, especially since, in one case, one studied the Sox4 role in isolation (reference 24), whereas in the other case, one looks at all Sox proteins. This contrasts with how the authors justify some of their claims by citing abstract (reference 37) or papers published in journals rarely cited.

We are sorry that our handling of the literature was perceived as inappropriate or incomplete. This was not intentional by any means. We selected references based on the value of the studies, not based on journal type. Of note, no abstract is included among the references.

The study led by Dr. Gautvik (Nissen-Meyer et al., 2007) examined the impact of a global heterozygous inactivation of *Sox4* on the bone phenotype of adult mice. Their cortical bone and trabecular bone findings are consistent with ours in SOXC conditional mutant mice. However, their molecular analyses showed reduced expression of osteoblast differentiation genes, including *Sp7*, *Bglap* and *Alpl*, but not *Runx2*, whereas ours showed upregulation of some of these genes in SOXC mutant MSCs and osteoprogenitor cells. The Gautvik group generated these data using primary osteoblasts from neonatal mouse calvaria, whereas we generated our data with cells isolated from adult mouse bone marrow. Major differences in experimental approaches, including the ages of the mice, bone origin of the cells, the cell types and culture conditions, likely explain differences in findings. This is now discussed in the Discussion. In the second work that we cited together with the Gautvik study (Gadi et al., 2013), SOX11 upregulated *Runx2* and *Sp7* expression and improved the differentiation of primary calvaria cells and osteoblastic cell lines *in vitro*. Since both studies used *in vitro* models of osteoblast differentiation to reach similar conclusions, and since the SOX4 and SOX11 proteins have highly similar functional domains, we described these data together in our original manuscript. In the revised manuscript, we have edited the text in the Introduction and Discussion to describe each published work separately and how our study consolidates or improves published knowledge. We also discuss implications of inactivating *Sox4* alone or the three SOXC genes simultaneously. This point does not strongly affect our conclusions since *Sox4* is the only SOXC gene expressed in adult osteogenic cells and since knockout of *Sox4* in the *OsxCre* and *Prx1Cre* lineages phenocopied the SOXC mutant mice.

Overall, and this applies to all figures, there are too many panels per figure, some with positive others with negative data. If anything, this removes some weight to the quality of the paper and some of these data could easily find their home in supplemental figures. This way of presenting is indeed overwhelming but is also less convincing than it could be anything it hurts the paper.

We acknowledge that most of our figures were very large. We have now revised the Results section thoroughly and reorganized the figures in order to make the manuscript easier to read and move data that are less central to the main messages to supplementary figures. We are not sure which data the reviewer considered negative, but we hope that the streamlining of the manuscript is helpful and addresses the reviewers' concern.

Do we really need to see cortical thickness represented in two different ways in Figure 1, for instance? This artificially takes a lot of space and leaves unaddressed the more important question of why the mutant mice die perinatally. Given the nature of the Cre used in this study, this perinatal lethality, and low body weight at weaning, one is surprised by the absence of any histological or molecular analysis during development or in the first weeks of life.

We assume that the reviewer referred to microCT images and quantification data as the two different ways of documenting cortical thickness. We originally showed both in a main figure because it is common in the field to show these complementary data. After weighing the pros and cons of moving images to supplementary figures, we opted to keep the microCT images of trabeculae in long bones in the main figure (Fig. 2a), because the trabecular phenotype is one of the most novel and striking findings of our study and visualizing it is certainly helpful. In contrast, we moved the images of cortical bones to a supplementary figure (Fig. S2a) because this phenotype is less striking and was not investigated in as much depth at the molecular level as the trabecular phenotype.

Mutant mice died only occasionally when they were raised without doxycycline-supplemented food, thus when *OsxCre* was active from embryonic stages. We agree that it could be interesting to determine the cause(s) of the neonatal defects of these mice, but we believe that this investigation would best be pursued in a separate study, given that the present one focuses on the adult roles of SOXC in bone and is already very large. We hope that the reviewer will agree with us. We speculate that this phenotype reflects some of the essential roles of SOXC previously described in many developmental processes (Penzo-Mendez, *Int J Biochem Cell Biol*, 2010). *OsxCre* has indeed been shown to be active in several cell types other than osteoblasts, including stromal, perivascular and adipocytic cells of the bone marrow, olfactory glomerular cells, gastric and intestinal epithelial cells, and brain cells (Chen et al., *PLoS One* 2014; Davey et al., *Transgenic Res* 2012). More recently it was also found to be active in hematopoietic cells (Ricci et al., *Elife* 2020) and peripheral adipocytes (Davis et al., *Sci Rep* 2022).

If we take a step back there is no need to have one figure for cortical bone, one figure for trabecular bone analyzed by microCT, and another one for trabecular bone analyzed by classical histological methods. There is instead a need for a more synthetic presentation. This is even more important since the author failed to identify a mechanism for the more severe phenotype in female than in male mice. In contrast, the demonstration that among the Sox protein, only Sox4 regulates trabecular bone mass is convincingly done in a single figure, as it should.

We have reorganized the figures to have more synthetic presentations of the work. Figure 1 now shows the main microCT and histomorphometry data for cortical bone and Figure S2 shows complementary data. Figure 2 now shows the main microCT and histomorphometry data for trabecular bone and Figure S3 shows complementary data. Likewise, we have also reorganized several other figures.

The last part of this study includes a transcriptomic analysis performed at 7 or 13 weeks of age in bone marrow mesenchymal stem cells of control and mice lacking all Sox proteins. In view of

the bone phenotype of the mutant mice, increased bone formation, and decreased bone resorption this analysis does not provide clear answers to explain why this dual histological phenotype take place.

Our transcriptomic analysis shows that SOXC inactivation significantly alters the expression of numerous secreted factors and cytokines that have been shown in previous studies to be involved in osteoblastogenesis and osteoclastogenesis. In particular, genes downregulated in mutant cells encode multiple well-known anti-osteoblastogenic and osteoclastogenic factors, and vice versa, upregulated genes encode many osteoblastogenic and anti-osteoclastogenic factors. Thus, these data provide highly plausible explanations for the increase in bone formation and decrease in bone resorption occurring in the trabecular bone of SOXC mutant mice. We have revised the manuscript, including the graphical summary presented as Fig. 7 and its legend to try and make the key findings of our study as clear as possible.

Here again, figures are cluttered with negative results that add little to the story. Why is bone formation decreased in the mutant mice? Does Sox4 regulate Type II collagen gene expression in a more direct measure than the one presented? Do Sox4 or does Sox4 alone regulate the expression of the interferon genes in osteoblasts in culture? How is the expression of Ank and alkaline phosphatase in the mutant bone and osteoblasts? What does the author mean by “Sox4 proteins generate a bone milieu delaying osteoblast differentiation but favoring bone mineralization? Rather, it would have been more important to perform in vivo experiments supporting a role of interferon in the ontogeny of this phenotype.

We are sorry, but it is unclear to us which data are considered by the reviewer to be negative. Our study includes many controls (such as experiments testing the effect of *OsxCre* itself and DOX food itself), mice of both sexes and at increasing ages, and several bone types. We believe that all data are important, even if some of the conditions tested did not yield major effects. They are important for scientific rigor (for instance, to verify that the effects seen upon SOXC inactivation are not due to the *OsxCre* transgene itself). They are also important to demonstrate that the actions of SOXC are dependent upon the context, including sex, age and bone type. We have moved several control experiments to supplementary figures. We have also tried to streamline the manuscript as much as possible. We hope that we have addressed this reviewer’s comment adequately.

The dynamic histomorphometry data of cortical bone that we have now added to our revised manuscript show that the bone formation rate (BFR/BS) is significantly reduced in *SOXC^{OsxCre}* mice at 5 weeks, but not at older ages (Fig. 1d,e). In contrast, the dynamic histomorphometry data of trabecular bone show that BFR/BS is increased in mutant females and males reaching adulthood (Fig. 2g and S4g). This increase matches the increase in osteoblast numbers detected at that age and fits with evidence from our transcriptomic analysis that SOXC inactivation facilitates osteoblast differentiation from mesenchymal progenitor cells. It will be worth in a future study to investigate the mechanism whereby SOXC inactivation lessens bone formation in juvenile mouse cortex.

Did the reviewer mean to say type I collagen (osteoblast marker) rather than type II collagen (chondrocyte marker)? While *Col2a1* (type II collagen gene) was identified among the top downregulated genes in several SOXC mutant clusters (see Table S2), it was not identified in the SOX4 or SOX9 regulon by SCENIC (see Table S4). The meaning of its downregulation is unclear since the importance of *Col2a1* in MSCs and osteoblastic cells is unknown. We therefore did not comment on this gene and many others in our study, having chosen to focus on differentially expressed genes whose contributions to bone formation and bone mass control are known.

The *Ank* gene is not differentially expressed in osteogenic cells of *SOXC^{OsxCre}* females at 7 or 13 weeks of age, whereas *Alpl* is upregulated in *Sca1⁺* cells (log2FC = -0.961) and downregulated in osteoblasts (log2FC = 0.281) at 13 weeks. These findings for *Alpl* support evidence that *SOXC* inactivation promotes osteoblastic differentiation of mesenchymal progenitor cells, but renders osteoblasts less efficient in mineralizing the bone matrix. However, since these changes are not among the most striking ones, we did not highlight them in the manuscript (which is already extremely large), but the data can be found in Table S2.

We have deleted the sentence referring to the bone milieu. This sentence was used to draw a conclusion on the expression changes observed in *SOXC* mutant cells for a number of secreted factors known to control osteoblast differentiation and bone mineralization. The bone milieu was referring to the extracellular environment in which these factors are secreted.

In revising the Results section on the analyses of molecular pathways and regulons affected in *SOXC* mutant cells, we opted to describe the data related to interferon signaling together with the data related to bone formation, ossification and wound healing. We better explain in Results and Discussion the possible meaning of the identification of interferon signaling-related pathways by GO and SCENIC analyses. We hope that this new presentation is clear and that the reviewer will agree that performing *in vivo* experiments to support a role for interferon in generating the bone phenotype of *SOXC* mutant mice is beyond the scope of the present study.

Reviewer #1 (Remarks to the Author):

The authors have addressed most of my concerns. I have only a few remaining suggestions:

1. To definitively prove that the C1 population in scRNA-seq indeed corresponds to bone marrow but not periosteal PaS cells (both of which express *Pdgfra* and *Ly6a*), they should at least add *Pdgfra* into the dot plots (Figures 3C and 4C). Also, it would be nice to perform qPCR on flow cytometrically sorted Lin-*Pdgfra*+*Sca1*+ vs. Lin-*Pdgfra*+*Sca1*- cells from the flushed bone marrow to show that *Col3a1* and *Postn* are enriched in bone marrow PaS cells. Cultured BMSCs dramatically change their expression profile as compared to primary uncultured BMSCs.
2. The modified Figure S2 is more acceptable now. Please avoid drawing strong claims on pseudotime analyses, which only infers but not definitively proves the lineage relationships among different populations.
3. Cell-cell interaction analysis such as CellPhoneDB is recommended to further strengthen the upstream regulatory mechanisms (Minor point #4).

Reviewer #2 (Remarks to the Author):

The authors have adequately addressed the reviewers comments and have presented a more concise story.

Dear Dr. Pattison and Reviewers,

Thank you for reconsidering our manuscript for publication in Nature Communications. We were very happy that both reviewers appreciated most of our revisions, which were made according to their excellent suggestions. We understand that Reviewer 1 had a few remaining suggestions to try and further improve our manuscript. We have now addressed these suggestions as best as we could, as explained below. We show the Reviewers' comments in black and our responses in blue. Please also see our revised manuscript file and revised Figures 3 and 4.

We hope that you will appreciate our responses and manuscript modifications and that you will find the manuscript now ready for acceptance.

Yours sincerely,

Véronique Lefebvre and Marco Angelozzi

Reviewer #1 (Remarks to the Author):

The authors have addressed most of my concerns. I have only a few remaining suggestions:

1. To definitively prove that the C1 population in scRNA-seq indeed corresponds to bone marrow but not periosteal PaS cells (both of which express *Pdgfra* and *Ly6a*), they should at least add *Pdgfra* into the dot plots (Figures 3C and 4C). Also, it would be nice to perform qPCR on flow cytometrically sorted Lin-*Pdgfra*+*Sca1*+ vs. Lin-*Pdgfra*+*Sca1*- cells from the flushed bone marrow to show that *Col3a1* and *Postn* are enriched in bone marrow PaS cells. Cultured BMSCs dramatically change their expression profile as compared to primary uncultured BMSCs.

As suggested by the reviewer we have now added *Pdgfra* to the dot plot of cell markers in Figures 3C and 4C. We have also added *Col3a1* to the dot plot in Figure 3C to show that *Sca1*⁺ cells in the bone marrow dataset from Zhong et al., *ELife* 2021 highly express *Col3a1* too. However, as the reviewer pointed out, since both periosteal and bone marrow *Sca1*⁺ cells express *Pdgfra*, this marker is not useful to definitively distinguish the two populations.

Additionally, we performed an RNA in situ hybridization (RISH) assay for *Col3a1* on sections from adult mouse tibia, and RT-qPCR assays for *Sca1*, *Col3a1* and *Postn* on Lin⁻CD31⁻SCA1⁺ cells flow cytometrically sorted from the flushed bone marrow or the periosteum. As shown in the figure below, the RISH assay (panel A) shows high expression of *Col3a1* in almost all periosteal cells, and most strongly in the cambium layer. *Col3a1* expression was also detected in cells lining bone trabeculae and in the bone marrow. RT-qPCR assays (panel B) showed that the expression of *Ly6a* (encoding SCA1) and *Col3a1* was enriched in both periosteal and bone marrow Lin⁻CD31⁻SCA1⁺ cells with respect to Lin⁻CD31⁻SCA1⁻ cells. Of note, *Postn* is not a good marker of these cells since it is expressed in other populations including osteoprogenitors and preosteoblasts as indicated by scRNA-seq assays. RT-qPCR assays showed a slight enrichment in bone marrow Lin⁻CD31⁻SCA1⁺ cells, but none in periosteal Lin⁻CD31⁻SCA1⁺ cells. Although these data support the existence of *Sca1*⁺*Col3a1*⁺ cells in the bone marrow, they do not rule out that the C1 cluster in our scRNA-seq dataset might contain residual periosteal cells in addition to bone marrow cells. Thus, we did not incorporate these data in our manuscript.

We also looked in the literature (Duchamp de Lageneste et al., *Nat Commun* 2018; Matthews et al., *Elife* 2021; Jeffery et al., *Cell Stem Cell* 2022; Xu et al., *Bone Res* 2022) for clear markers of periosteal versus bone marrow *Sca1*⁺ MSCs but could not identify any. Future investigations

dedicated to the identification of these markers will be needed, but we believe that this is beyond the scope of our study. Rather, to address the Reviewer's valuable point, we have now modified the sentence regarding the C1 cluster in the Results section as follows: "C1 contained *Sca1*⁺ (also referred to as *Ly6a*⁺) MSCs originating from bone marrow (also known as PDGFRA⁺SCA1⁺ or PαS MSCs⁴² and as early mesenchymal progenitors or EMPs³⁵) and possibly also from residual periosteum⁴³. We hope that the Reviewer will approve our response.

Figure legend. (a) *Col3a1* RNA in situ hybridization of a section through the proximal tibia of a 13-week-old mouse. High-magnification images of highlighted areas are shown at the bottom: a', periosteum and cortical bone; b', trabecular bone region; c', bone marrow. The magenta color represents RNA signal. The blue color resulting from counterstaining with hematoxylin was desaturated (to grey color) using Adobe Photoshop. AC, articular cartilage; BM, bone marrow; CB, cortical bone; GP, growth plate; P, periosteum; POC, primary ossification center; SOC, secondary ossification center; TB, trabecular bone. (b) RT-qPCR assays of the expression levels of *Ly6a*, *Col3a1* and *Postn* in Lin⁻CD31⁻SCA1⁻ (grey bars) and Lin⁻CD31⁻SCA1⁺ (black bars) cells isolated separately from the periosteum and bone marrow of a 13-week-old mouse. Red numbers on top of the black bars indicate the expression fold changes between SCA1⁺ and SCA1⁻ cells.

2. The modified Figure S2 is more acceptable now. Please avoid drawing strong claims on pseudotime analyses, which only infers but not definitively proves the lineage relationships among different populations.

We agree with the reviewer that pseudotime analyses have suggestive rather than conclusive power in describing cell lineage relationships and trajectories. Indeed, taking in account this consideration, we carefully used the verbs "suggest" and "predict" and the conditional auxiliary verbs "would" and "might" when describing these analyses. We have now further addressed this concern by modifying the sentence describing the RNA velocity analysis as follows: "In agreement with Slingshot, RNA velocity analysis, which predicts the step-by-step directionality of cell progression, also suggested a faster progression of SOXC mutant MSCs/osteoprogenitors towards OB differentiation (Fig. 6b)".

Additionally, we would like to stress the fact that predictions obtained by pseudotime analyses are supported by other data, already published or generated in this manuscript. In particular:

- the notion that LEPR⁺ and SCA1⁺ MSCs are osteoblast progenitors is not new and has been largely investigated in the literature (Morikawa et al., J Exp Med 2009; Zhou et al., Cell Stem Cell 2014; Zhong et al., Elife 2020; Ambrosi et al. Elife 2021).
- the notion that SOXC mutant MSCs progress faster towards osteoblast differentiation is supported by our in vitro differentiation assays and by in vivo molecular changes that we identified by scRNA-seq. Also, this notion aligns well with the increased number of osteoblasts and trabeculae observed in mutant mice.

3. Cell-cell interaction analysis such as CellPhoneDB is recommended to further strengthen the upstream regulatory mechanisms (Minor point #4)

We thank the reviewer for this suggestion. Most of the cytokines and secreted factors that mediate cross-talks among cells and are differentially expressed in SOXC mutant cells are expected to affect osteoclasts and their precursors. Since these cells are not represented in our scRNA-seq dataset (we sorted out hematopoietic cells, including myeloid lineage cells), the most interesting cell-cell interactions would likely be absent in this analysis. Additionally, the roles of these cytokines on osteoblast and osteoclast populations have been previously documented in several other studies which are cited throughout the Results and Discussion sections. Thus, we concluded that performing this analysis would not be very informative. We hope that the reviewer will agree with our rationale for not performing this analysis.

Reviewer #2 (Remarks to the Author):

The authors have adequately addressed the reviewers comments and have presented a more concise story.

We are glad that the reviewer appreciated our revised manuscript.

Reviewer #1 (Remarks to the Author):

The manuscript is ready to be accepted for publication. No further questions or comments.